# Cross-environment Cooperation Enables Zero-shot Multi-agent Coordination

**Kunal Jha** [1]  **Wilka Carvalho** [2]  **Yancheng Liang** [1]  **Simon S. Du** [1]  **Max Kleiman-Weiner** [* 1]  **Natasha Jaques** [* 1]

## Abstract

Zero-shot coordination (ZSC), the ability to adapt to a new partner in a cooperative task, is a critical component of human-compatible AI. While prior work has focused on training agents to cooperate on a single task, these specialized models do not generalize to new tasks, even if they are highly similar. Here, we study how reinforcement learning on a **distribution of environments with a single partner** enables learning general cooperative skills that support ZSC with **many new partners on many new problems**. We introduce *two* Jax-based, procedural generators that create billions of solvable coordination challenges. We develop a new paradigm called **Cross-Environment Cooperation (CEC)**, and show that it outperforms competitive baselines quantitatively and qualitatively when collaborating with real people. Our findings suggest that learning to collaborate across many unique scenarios encourages agents to develop general norms, which prove effective for collaboration with different partners. Together, our results suggest a new route toward designing generalist cooperative agents capable of interacting with humans without requiring human data. Code for environment, training, and testing scripts and more can be found at https://kjha02.github.io/publication/cross-env-coop.

## 1. Introduction

Humans excel at ad-hoc cooperation, readily adapting to new partners and environments by attending jointly to relevant objects, reasoning about shared intentions, and playing their role within an implicit collective plan (Tomasello, 1999; Kleiman-Weiner et al., 2016; Shum* et al., 2019; Wu* et al., 2021). This ability to effectively represent collective tasks allows cooperation to transfer across both partners and environments. For instance, after mastering a family recipe with their parents, a novice chef can easily cook the same dish (and much more) at home with their spouse. These cognitive mechanisms may be important for building AI that coordinates in novel scenarios. However, current reinforcement learning (RL) methods have yet to address this challenge (Wang et al., 2024). Cooperation across partners on a single problem has been studied, but agents that can zero-shot coordinate (ZSC) with new partners in unfamiliar environments will unlock flexible, human-compatible AI agents in a range of applications: household robots, adaptive educational assistants, or autonomous vehicles (Ma et al., 2023; Stone et al., 2010b; Atchley et al., 2024; Dinneweth et al., 2022; Ribeiro et al., 2021).

Prior work on ZSC has mainly focused on the challenge of adapting to novel partners, including novel human partners. Typical RL approaches use methods like population-based training (PBT) and variations on self-play (SP), approaches which helped algorithms such as AlphaStar achieve superhuman performance in zero-sum games (Vinyals et al., 2019). These approaches work by leveraging "partner diversity," during training time. They simulate diverse training partners with the idea that these varied experiences will be sufficient for agents to generalize to human partners (Yan et al., 2023; Carroll et al., 2020; Strouse et al., 2018; Sarkar et al., 2023). Although agents trained under this paradigm often do adapt to new partners, this adaptation is limited to the single environment they are trained on. PBT, as it is often deployed (and similar approaches), fails to generalize when faced with even a slight variation of the same problem. Thus, current PBT-based approaches must be retrained every time there is a slight variation in the environment or task. This is not a recipe which can scale well to the real world (Yan et al., 2023). If agents can only successfully cooperate with others on the specific environment they were trained on, such as a household robot which can only interact with people in one bedroom rather than the entire house, they lack a more general notion of cooperation — the ability to flexibly interact in a broad range of scenarios with many people.

In this work, we investigate the following question: how can

---

*Equal contribution  [1]Department of Computer Science, University of Washington, Seattle, WA  [2]Kempner Institute for the Study of Natural and Artificial Intelligence, Harvard University, Cambridge, Massachusetts. Correspondence to: Kunal Jha <kjha@uw.edu>.

*Proceedings of the 42nd International Conference on Machine Learning*, Vancouver, Canada. PMLR 267, 2025. Copyright 2025 by the author(s).

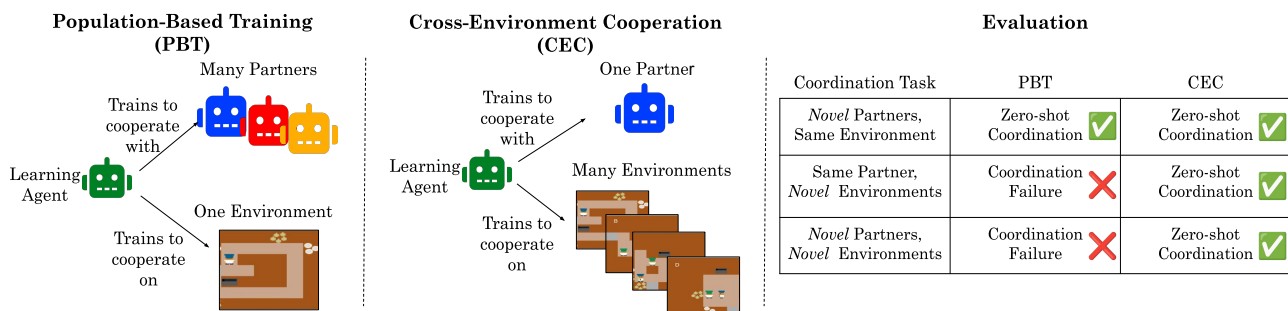

*Figure 1.* Overview of learning general coordination through Cross-environment Cooperation (CEC). By training agents in self-play on a large distribution of environments, we find that agents develop the ability to coordinate with novel partners and novel problems, contrasting with prior work which suggests self-play is insufficient for learning general norms for cooperation.

we train AI agents capable of zero-shot collaboration with *novel partners* in *novel tasks*? To investigate this, we focus on two main sources of variation during training: partner diversity and environment diversity. While PBT methods focus on partner diversity to encourage agents to adapt to different strategies on a single-task, we hypothesize that training across diverse environments will instead foster a richer understanding of the task structure itself, enabling agents to generalize to novel partners and settings.

To systematically test our hypothesis and isolate the impact of environment variation compared to partner variation, we present a new paradigm, **Cross-environment Cooperation (CEC)**, in which agents are trained via self-play (with a single partner) to cooperate across a wide variety of procedurally generated tasks. To evaluate CEC, we first test on both a toy environment to illustrate key principles, and then scale up to the human-AI coordination benchmark, Overcooked. Overcooked is a 2-player, cooperative 2D cooking game where an AI agent collaborates with an AI or human partner to prepare a recipe (Carroll et al., 2020; Wu* et al., 2021; Strouse et al., 2022; Zhao et al., 2022; Sarkar et al., 2023). We design a procedurally generated Overcooked environment implemented in Jax, enabling high performance and efficient training speeds of up to *10 million steps per minute on a single GPU*. Unlike previous work, which studies at most five handcrafted Overcooked levels (Carroll et al., 2020; Strouse et al., 2018; Zhao et al., 2021; Myers et al., 2025), our generator creates up to $1.16 \times 10^{17}$ diverse and solvable kitchen configurations.

Our experiments intriguingly reveal that by training on diverse *environments*, agents learn to consistently improve generalization to new *partners* (see Figure 1). We conduct extensive simulated and human experiments to evaluate the performance of CEC agents against state-of-the-art (SOTA) baselines. Our human study reveals that CEC agents outperform PBT on performance and outperform all methods on subjective measures of cooperation, suggesting that the generalized cooperative skills learned through diverse envi-

ronmental exposure translate well to human-AI interactions.

The main contributions of this work are:

1. **Cross-Environment Cooperation:** A paradigm for ZSC that replaces partner diversity with environment diversity. Via procedural environment generation, we eliminate the need for partner populations while training a single policy to cooperate across diverse tasks.

2. **Algorithmic infrastructure:** (1) Fast Jax-based procedural generation for Overcooked ($1.16 \times 10^{17}$ layouts, 10M steps/min); (2) A toy benchmark isolating environmental vs. partner diversity, demonstrating CEC's partner and environment generalization over PBT.

3. We find that **environment diversity outperforms partner diversity in cross-play performance**. Through analyzing generalization to different agents in cross-play, and via Empirical Game-Theoretic analysis, we show CEC outperforms PBT and other baselines.

4. **Human-AI cooperation insights:** Human studies reveal CEC outperforms PBT in cooperation score and state-of-the-art methods in qualitative evaluations by participants, demonstrating its success in realistic cooperative scenarios.

## 2. Related Work

Building agents that can coordinate with novel people quickly has the potential for broad impacts across robotics (Breazeal et al., 2005; Sheridan, 2016), digital assistants (Guo et al., 2024; Poddar et al., 2024; Ying et al., 2024), and other scientific domains (Ghazimirsaeid et al., 2023; Roche et al., 2008; Castelfranchi, 2001).

**Self-Play (SP)** has been highly successful in zero-sum games because there is only a single mixed strategy equilibrium (Xia et al., 2018; Silver et al., 2017; Vinyals et al., 2019; Zhou et al., 2020). However, when applied to cooperative games self-play often converges to a brittle and inflexible policy that struggles with unfamiliar partners that

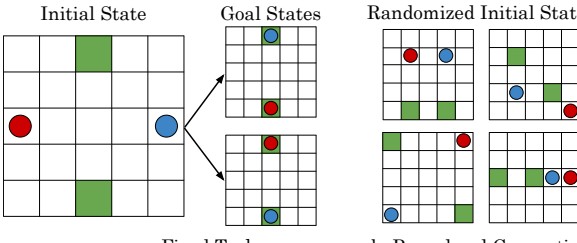

*Figure 2.* The Dual Destination Problem. In the fixed task (a), players start in opposite squares and must enter different green squares from each other to receive a reward. In the procedurally generated variation (b), the initial positions of the green goal cells and agents are randomized.

play in novel ways (Strouse et al., 2022; Ma et al., 2023). Intuitively, when training in self-play in a zero-sum game, the training procedure provides a continuous curriculum of diverse experience as the model is continuously rewarded for exploiting their opponent. In cooperative games with multiple equilibria, the situation is reversed. Once an equilibrium is found, neither player has an incentive to unilaterally explore other equilibria. Thus, diverse experiences are not generated through self-play in this context.

**Population-based training (PBT)** has been explored as a way to combat the shortcomings of self-play in cooperative games. PBT trains a cooperator agent to learn a best response to a diverse pool of partners and thus experience multiple equilibria of the game (Vinyals et al., 2019). PBT-based methods generally outperform self-play in zero-shot Human-AI coordination. In Overcooked, previous works (Carroll et al., 2020; Strouse et al., 2022; Zhao et al., 2022; Sarkar et al., 2023) induced partner diversity at training time through variations in neural network initializations or auxiliary loss functions. While PBT allows agents to train against different strategies for a single task, it is computationally expensive since each task requires first training a sizable population of agents. In contrast, our approach systematically varies the environment instead of the partner distribution, eliminating the need to train multiple policies.

**Procedural Environment Generation.** Recent work has demonstrated that procedurally generated environments can improve the generalization of reinforcement learning (RL) methods in single and multi-agent settings (Cobbe et al., 2020; 2019; Fontaine et al., 2021; Carion et al., 2019; Chen et al., 2023; Samvelyan et al., 2023b). These studies show that exposure to a large and diverse set of samples enhances generalization (Cobbe et al., 2020). However, they typically evaluate agents with the same team seen during training, which doesn't address the core challenge of zero-shot coordination (ZSC). There has been work in the zero-sum setting showing environment diversity and partner diversity are not completely orthogonal axes and prioritizing either

can be suboptimal, and that regret-based sampling enables agents to efficiently learn to compete in novel scenarios with novel partners (Samvelyan et al., 2023a; 2024). McKee et al. (2022) find environment diversity improves agents' ability to collaborate on novel levels in the ZSC setting, but they do not demonstrate the benefits of this approach with ad-hoc partners such as humans, the effects of combining partner with environment diversity, the relative benefits of environment diversity compared to partner diversity, or a deeper qualitative understanding of the effect of environment diversity on learned cooperation strategies. Related to our work, Ruhdorfer et al. (2024) study unsupervised environment design (UED) in the context of Overcooked. However, their work does not prevent the generation of impossible coordination challenges (Dennis et al., 2021; Mediratta et al., 2023; Li et al., 2023) and their results reveal poor generalization performance to new partners on held-out levels. In contrast, we show training across many coordination tasks with the same partner still can improve generalization to novel partners.

## 3. Technical Preliminaries

A two-player cooperative Markov Game is defined as a tuple $\langle S, A, T, R, H \rangle$, where $S$ are states, $A$ are actions (shared by both agents), $\mathcal{T}$ is the deterministic function $\mathcal{T} : S \times A \times A \rightarrow S$, rewards are $R : S \times A \times A \rightarrow \mathbb{R}$, and the game horizon is $H$. Both the transition and reward functions are assumed to be unknown. We investigate the problem of training a cooperator policy $\pi_C$ which can achieve high reward when paired with many different partner policies $\pi_p \sim \mathcal{P}$, where $\mathcal{P}$ represents a distribution of possible partners, analogous to the distribution of humans an agent might need to assist. We assume that cooperation also takes place across many different environments. Here we define each environment, or cooperation task, as drawn from a set of possible tasks $m \sim \mathcal{M}$. The task $m$ defines the initial state distribution, $p(s_0|m)$, but tasks share transition dynamics $\mathcal{T}$ and reward function $R$. Each different task $m$ is analogous to a new environment layout that may significantly alter the space of effective coordination strategies that will lead to high reward. We define the score obtained in the cooperative game obtained by summing the joint per-timestep rewards when the cooperator $\pi_C$ plays partner $\pi_p$ in task $m$ as:

$$S(\pi_p, \pi_C, m) = \mathop{\mathbb{E}}_{\substack{s_0 \sim m, s \sim \mathcal{T} \\ a^P \sim \pi_p, a^C \sim \pi_C}} \left[ \sum_{t=0}^{H} R(s_t, a_t^p, a_t^C) \right]$$

**Cross-Play (XP) Evaluation:** The objective of cross-play (XP) evaluation is to test how well a cooperator policy $\pi_C$ performs when paired with a novel partner policy $\pi_p$ in an environment $m$, i.e. evaluate $S(\pi_p, \pi_C, m)$. In line with prior work (Strouse et al., 2022; Zhao et al., 2021; Yan

et al., 2023), we simulate novel partners for XP evaluation by training multiple agents using the same algorithm with different initial random seeds. Doing so results in different network initializations and exploration patterns, causing agents to learn arbitrarily different, yet successful, ways to coordinate on a problem (Carroll et al., 2020). Collaborating with novel initializations of the same learning algorithm is referred to as the Zero-shot Coordination (ZSC) setting (Hu et al., 2020). We evaluate all agents in ZSC, but also study cooperation in settings where agents must collaborate with novel partners that trained with different learning algorithms, also referred to as "Ad-hoc Teamplay" (Stone et al., 2010a).

**Population-based training (PBT):** (e.g. (Strouse et al., 2022; Zhao et al., 2021; Liang et al., 2024)) aim to construct a diverse set of simulated partner policies $P = \{\pi_1, \pi_2, \ldots, \pi_n\}$ to simulate various human behaviors, but focus training on a single task $m$. Thus, they optimize:

$$J(\pi_C) = \mathbb{E}_{\pi_i \sim P}[S(\pi_i, \pi_C, m)] \qquad (1)$$

For example, Fictitious Co-Play (FCP) is a popular PBT-based method, where a single cooperator agent tries to learn a best-response policy to a population of self-play agents at different points of their learning progress (Strouse et al., 2022).

## 4. Cross-Environment Cooperation Improves Partner Generalization

The core hypothesis in our work is that enhancing the diversity of training tasks will improve agents' ability to generalize to both new tasks, and new cooperation partners. Our approach, *Cross-Environment Cooperation (CEC)*, trains a cooperator policy, $\pi_C$, by sampling diverse tasks from a broad distribution of tasks, $m \sim \mathcal{M}$. To isolate the effect of task diversity from partner diversity, we use self-play to train the CEC cooperator policy. The objective of CEC is thus defined as:

$$J(\pi_C) = \mathbb{E}_{m_i \sim \mathcal{M}}[S(\pi_C, \pi_C, m_i)] \qquad (2)$$

By sampling diverse tasks, CEC encourages the cooperator policy to generalize its behavior to handle a variety of cooperative scenarios. Unlike PBT, which requires maintaining a large pool of partner policies, CEC simplifies the training process because it only requires training a single policy. We investigate whether using CEC to optimize Eq. 2 can actually provide ZSC performance gains over using PBT to optimize Eq. 1.

**Dual Destination Game.** To gain intuition for the existing gap in the ability of multi-agent reinforcement learning methods to collaborate with partners they have not seen during training (Lowe et al., 2017), we design a simple gridworld environment as follows: two agents (red and

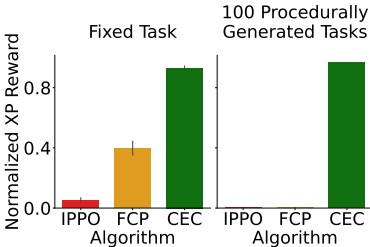

*Figure 3.* Evaluation of IPPO and FCP baselines on the Fixed and Procedurally generated versions of the Dual Destination problem (error bars show the standard error of the mean). CEC generalizes better in both cases ($p < 0.001$ for t-tests comparing CEC to both FCP and IPPO).

blue) begin on the opposing sides of a grid without any walls. They are both given a $+3$ reward for moving to opposing green grid cells. In the basic setup, these grid cells are equidistant from each other and the agents' original starting position, as illustrated in Figure 2(a). The agents receive a $-1$ step cost and can move up, down, left, right, or stay in place. The state is fully observable.

We hypothesize that self-play (SP) agents will have poor cross-play (XP) performance in this game, because they will pick a convention (red agent goes up, blue goes down) that will not generalize to different SP agents which found a different convention. PBT methods like FCP may learn a more general strategy, such as "go up if someone goes down, and down if they go up." However, these policies are still brittle: if the goal location is shifted by even a few squares or the agents begin in a novel state, the FCP-trained agent will not have experience training in this environment, and thus, we hypothesize that it will fail.

To test whether cross-environment training can ameliorate these shortcomings, we train a CEC model on the Dual Destination game by randomizing the agent starting and goal positions at the beginning of a new episode. We show examples of potential initial states in Figure 2(b). We ensure that the train task used for the other methods ('Fixed Task') is *held out* from the CEC train distribution. We then use IPPO, a SOTA multi-agent SP algorithm (de Witt et al., 2020), to train agents to optimize the CEC objective in Eq. 2. The CEC model was trained without a population of other agents or any additional auxiliary losses.

**AI-AI ZSC Evaluation.** We compare the ZSC performance between SP (IPPO), FCP, and CEC in the Dual Destination tasks shown in Figure 2, and include the task used to train the SP and FCP methods ('Fixed Task'), as well as 100 novel procedurally generated tasks. We first measure how well models generalize cooperation to novel partners (cross-play performance with novel partners). Second, we measure how well they generalize to novel partners in novel

environments (XP performance with novel partners in novel environments).

Figure 3 illustrates the pitfalls of training a naive cooperator in self-play: even with just two options to choose from in response to another's actions, agents cannot adapt well at test-time to a novel partner's strategy and thus fail to coordinate *even on the tasks they've seen during training*. FCP does better by learning to adapt to strategies within a single level by training against a diversity of partners. However, Figure 3 shows that it fails to complete similar tasks with minor structural differences. In contrast, CEC agents perform better with novel partners in the Dual Destination Game *on an evaluation layout it has never seen during training* (Figure 3). For context, **an oracle agent that perfectly responds** to the two optimal strategies in Figure 2 would require 3 steps of receiving a -1 step cost to move to the target location before receiving a (3 positive - 1 step cost) reward for 97 steps, equating to a **0.955 normalized reward. CEC scored 0.931 normalized reward** with a standard error of 0.013, indicating it underperforms the oracle's cross play performance by about 2.5%.Moreover, Figure 3 indicates that CEC models also generalize to 100 new initial state configurations *and* new partners *using the same amount of compute* as the vanilla single-task IPPO baseline. We extended our analysis in the Appendix to include partially observable and multi-task variations of the Dual Destination problem (Appendices A.2 and A.3, respectively). Our findings consistently show that even with imperfect information and task uncertainty, **CEC agents outperform population-based and naive self-play methods in cooperating with novel partners on novel problems**, mirroring the trends observed in the fully observable ZSC case.

These findings suggest that investing in training cooperative agents on a large distribution of environments is potentially much more effective than training across a large distribution of partners, just as prior work has suggested this kind of procedural generalization (or domain randomization) improves generalization in the single-agent setting in robotics (Jakobi, 1997; Sadeghi & Levine, 2016; Tobin et al., 2017).

**Procedurally Generated Overcooked.** To test whether these findings replicate in a more scaled-up scenario, we extend the Overcooked environment from the JaxMARL project to support a wider variety of levels (Rutherford et al., 2023).

Previous work on Overcooked has focused on five layouts that possess diverse coordination challenges (Carroll et al., 2020; Strouse et al., 2022; Zhao et al., 2021; Yan et al., 2023). For our new environment, we uniformly sample the wall structure from each of these five layouts, then randomly generate features like goals, plates, pots, and onions within the grid. Doing so structures the generation process and introduces variable complexity depending on where the

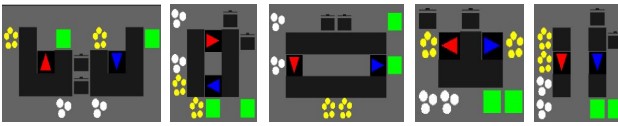

Figure 4. Five original Overcooked layouts. Left to right: *Asymmetric Advantages, Coordination Ring, Counter Circuit, Cramped Room, Forced Coordination*.

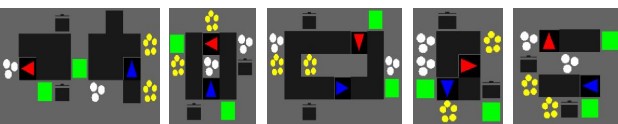

Figure 5. Sample from the billions of solvable, diverse Overcooked tasks created by our procedural environment generator.

objects are sampled. We ensure tasks remain solvable by placing essential objects in reachable areas from the layouts in Figure 4 and detailed in A.1. We introduce additional environment diversity by sampling additional goal locations, plate piles, pots, and onion piles on unoccupied walls. The grids are also rotated randomly to encourage better representation learning. For example, if an agent learned to complete a task on a very wide layout, it should be able to do that same task if it is rotated vertically by performing a simple mapping from the left/right actions to up/down, and vice versa.

Our procedural generator creates new coordination challenges in Overcooked, shown in Figure 5. Moreover, Jax allows us to run the entire training and evaluation pipeline, from the environment generation to the neural network updating of agents, at *10 million steps per minute on a single GPU*. We leverage this speed to train CEC agents, and all other baselines, for 3 billion steps. By standardizing the training duration for all algorithms, we can directly compare whether it is better to invest compute in training on more environments or in methods like PBT.

## 5. Experiments

From this point on, we will refer to agents trained to cooperate across multiple coordination challenges by optimizing Eq. 2 as CEC, agents trained on a single task as ST, the self-play performance of the agents as SP, and the cross-play performance of the agents as XP. Our primary focus is on the XP setting, which measures how well agents collaborate with novel partners on tasks they have never seen before.

When evaluating AI-AI and Human-AI coordination, we aim to answer the following research questions:

1. Is increasing environment diversity more effective than increasing partner diversity for ZSC?

2. Compared to ST methods, how well do CECs general-

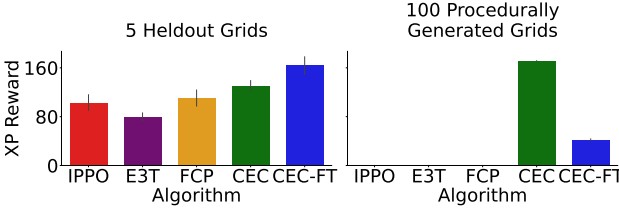

*Figure 6.* Evaluation of baselines on (left) 5 original Overcooked layouts vs. (right) 100 procedurally generated held-out layouts (standard error bars). Single-task methods and PBT struggle in both settings, while CEC agents generalize effectively. Finetuning CEC on a single grid improves XP performance on original layouts (outperforming FCP and IPPO; $p < 0.01$, t-test) but reduces generalization on novel layouts. CEC significantly surpasses all baselines in procedural generalization tasks ($p < 0.0001$, t-test).

ize cooperative strategies to novel environments?

3. How close can CEC come to state-of-the-art performance with novel humans for a single task it has never seen before through environment diversity alone?

**Baselines.** For single-task baselines, we train six seeds for three different agents for each of the original five layouts in Figure 4, one with vanilla IPPO (de Witt et al., 2020), one with FCP (Strouse et al., 2022), and one with Efficient End-to-End Training (E3T) (Yan et al., 2023). E3T is the state-of-the-art algorithm for single-task ZSC in Overcooked. It is a SP method that achieves high XP performance by adding randomness to one of the partner's policies during training, and training an auxiliary network which tries to predict the actions of other agents in the world. We test CEC's cross-play performance with novel partners on each of the single-tasks FCP and E3T were trained on.

**Evaluation Protocol.** Following prior work (Zhao et al., 2021; Carroll et al., 2020; Yan et al., 2023; Strouse et al., 2022), we first evaluate the ability of all models to generalize to new partners on the five original Overcooked Layouts 4. Note that we hold out those five layouts from the CEC generator, so that when we evaluate CEC on these layouts we are able to test generalization across both partners and tasks. Second, we introduce an additional evaluation setting where we have the Overcooked procedural environment generator create 100 coordination challenges that neither the ST baselines nor any of the CEC agents have seen during training and assess how well the different approaches can generalize to both novel partners and novel environments. We train six seeds for each type of agent. The architectural details are included in A.7.

**Cross-Environment Cooperation as a Pre-training Step.** Holding out the original five layouts from the CEC generator puts CEC at a disadvantage compared to ST baselines, which only have to generalize to novel partners, while CEC

has to generalize to new partners and environments. Towards mitigating the bias against our method and answering Research Question 3, we study the effects of trading off generalization for specialization in CEC models. After training a single CEC agent on a distribution of environments created by the procedural generator, we create five copies of the model, one corresponding to each of the original five layouts in Figure 4 which were held out during training. For each copy of the CEC agent, we perform an additional 100 million steps of training on a single layout with a reduced learning rate, again in self-play using IPPO. We call this approach "CEC-Finetune (CEC-FT)."

**Human Studies.** Due to the expense of human evaluations, our Human-AI evaluations only looked at the two most challenging grids of the original five, *Counter Circuit* and *Coordination Ring*. We recruit 80 human participants for our study using Prolific, where 40 participants complete each layout. Our study follows a protocol approved by our university's IRB. During the study, each user plays multiple rounds of Overcooked with a partner via a web interface, where in each round the partner is controlled by one of the models, in randomized order. We sample a new model trained with a different random seed for each of the game rounds. Subjects played with agents for 200 timesteps and the entire experiment took approximately 30 minutes. After each round, the user answered questions about their subjective experience playing with the agent using a Likert scale (Likert, 1932). This survey allows us to quantify the factors that most heavily influence overall performance and users' preferences for coordination partners. All of our Human-AI experiments and surveys utilized the NiceWebRL Python package (https://github.com/wcarvalho/nicewebrl), which leverages Jax's parallelizability to efficiently crowd-source participant data on reinforcement learning environments.

## 6. Results

**Q1: Is increasing environment diversity more effective than increasing partner diversity of ZSC?** Figure 6 shows the XP performance of the baseline models averaged across the five original Overcooked layouts. Although ST approaches are capable of adapting to partner strategies during training time, they fall short in XP compared to CEC models. Fine-tuning CEC agents on a single grid layout resulted in a substantial improvement in cross-play generalization performance. While it is unsurprising that training in the test environment helps performance, even without training in the test layout or using any strategies to enhance partner diversity, CEC outperforms all baselines in terms of XP. The success of CEC and CEC-Finetune in XP indicates the benefits of both broad pre-training with task-specific adaptation. More importantly, we obtain evidence to support an affirmative answer to research question

1: increasing environment diversity can actually be more effective at improving cross-*partner* generalization, than training on a single environment with many partners.

One challenge with assessing coordination using the single-task levels is that it does not control for the possibility that all seeds just happen to play the same strategy (e.g., "always rotate clockwise."). This would result in high cross play scores without the ability to generalize to novel partners. Therefore, we compute cross-algorithm cooperation scores, shown in Figures 21 and 22, to control for this possibility and evaluate agents in the Ad-hoc Teamplay setting. Using this cross-algorithm performance matrix to represent a meta-game, we conduct an **empirical game-theoretic analysis** where players choose model types rather than actions. Essentially, players in the meta-game choose a model (like CEC vs. FCP) in order to maximize cross-play score. This can also be see as treating Figures 21 and 22 as a payoff matrices, and analyzing the equilibria of these games. This approach allows us to interpret how a simulated population of agents would adopt different strategies over time, revealing attractors, equilibria, and cyclic behaviors that may not be apparent from simple win rates or Elo ratings (Wellman et al., 2024). Following the approach outlined in (Tuyls et al., 2018; Serrino* et al., 2019), Figure 8 shows the gradient of the replicator dynamic in these meta-games as a way of assessing the dynamics and equilibrium of the meta-game. For both the five original tasks and the 100 procedurally generated tasks, the direction of the gradient is towards either CEC or CEC-Finetune. This shows that CEC trained models generalize robustly across different algorithms.

**Q2: Compared to single-task methods, how well do CECs generalize their cooperative strategies to novel environments?** Similar to the Toy environment, the evaluation on the held-out Overcooked layouts created by the procedural environment generator, as depicted in Figure 6,

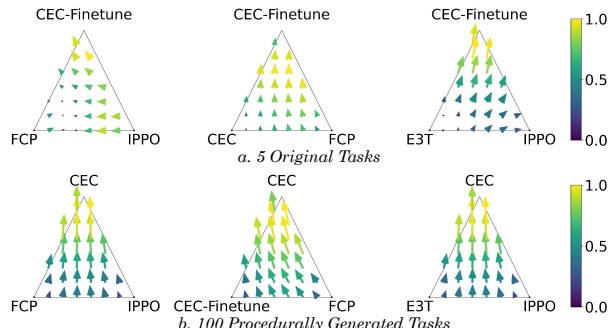

*Figure 8.* Empirical game-theoretic evaluation of cross-algorithm play on the (a) five original and (b) 100 procedurally generated Overcooked tasks. Arrows show the gradient of the replicator dynamic on the cross-algorithm meta-game. Vectors flow towards CEC and CEC-Finetune indicate they are likely equilibria.

reveals that FCP and E3T completely fail to generalize to novel tasks, receiving 0 reward on any of the 100 test tasks. While both methods try to increase partner strategy coverage in self-play by adding entropy to the partner policy, they fail to generalize to similar, procedurally generated layouts due to insufficient *environment coverage*. Without encountering the same task in diverse scenarios, agents learn low-level sequences (e.g., "move to the third cell from the left and interact") rather than higher-level task structures (e.g., "cook onions and deliver them"), hindering environment generalization.

We also assess how well humans can collaborate with ST single-task agents playing a layout they have never seen during training. We call the IPPO and FCP versions of these agents IPPO$^-$ and FCP$^-$ respectively. Just as in simulation, we find that models only optimized to cooperate on one task cannot generalize to new tasks with ad-hoc partners (Figure 9), and are consistently rated the most frustrating to play with by humans (Figure 27). Taken together with the FCP cross-play performance, this provides strong evidence that ST methods cannot generalize at all to novel, similar cooperative settings whereas CEC agents can.

Interestingly, Figure 6 shows CEC-Finetune did not perform as well as vanilla CEC on the novel procedurally generated layouts. This highlights a tension between generality and specialization in agent training. While CEC fine-tuning demonstrates that pre-training on diverse problems provides a solid foundation for learning to collaborate on a specific problem with new partners (Figure 6), it also results in a loss of the ability to generalize to novel levels.

In Appendix A.4, we explored whether combining partner and environmental diversity enhances agents' ZSC abilities. We trained E3T agents within the CEC framework and evaluated their XP performance on unseen tasks with unfamiliar

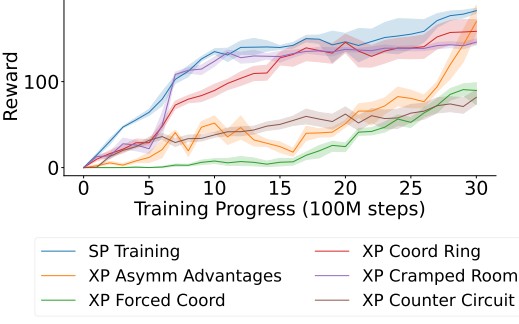

*Figure 7.* CEC SP Training Performance compared to XP Performance on 5 held-out levels. Despite the distribution sampling each layout predicate at uniform, CEC gets better at different layouts at different rates as it consistently improves across all tasks.

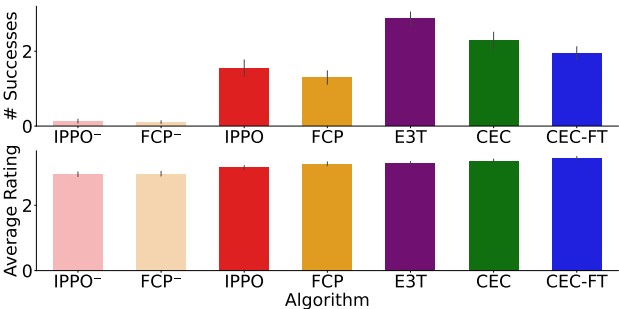

*Figure 9.* (Top) Average success rates of algorithms cooperating with ad-hoc human partners on *Counter Circuit* and *Coordination Ring*, with standard error bars. CEC outperforms PBT methods and approaches E3T's performance, despite only training on diverse layouts. Using a 2-sided t-test, CEC significantly outperforms FCP ($p < 0.001$, t-test). (Bottom) Human ratings of algorithms' cooperative ability across 7 metrics, averaged over *Counter Circuit* and *Coordination Ring* evaluations, with standard error bars. CEC and CEC-Finetune are preferred partners despite lower rewards. CEC-Finetune significantly outperforms FCP ($p < 0.01$, t-test) and E3T ($p < 0.01$, t-test).

partners. As Figure 16 illustrates, introducing novel partners while learning in a constantly changing environment *did not improve agents' ZSC capabilities*. This suggests further research is needed to determine how to best integrate these two forms of diversity to realize their combined benefits.

Last, we analyze XP performance at different points in the learning trajectory of CEC agents. Figure 7 shows the CEC SP training reward and the corresponding XP evaluation reward on each of the five layouts at the same point in training. We find that CEC XP performance improves at different rates across layouts. This can partly be explained by the diversity in optimal coordination strategies across the original five layouts (Figure 4). In both *Cramped Room* and *Coordination Ring*, the spatial constraints lead agents to discover cooperative strategies more quickly. However, for layouts such as *Counter Circuit*, where agents have many options for where to pass items along the middle border, whether to rotate around the middle wall clockwise or counter clockwise, etc., CEC's XP performance improves slower but increases steadily. It could potentially be driven even higher with more training.

**Q3: How close can CEC come to state-of-the-art performance with novel humans for a single task it has never seen before through environment diversity alone?** To answer this question, we analyzed how well CEC agents can collaborate with new people on novel tasks under the Ad-hoc Teamplay setting. As Figure 9 demonstrates, when evaluated in terms of score in the cooperation task, on levels that ST methods have seen during training, CEC and CEC-finetune are able to collaborate better than IPPO and

FCP models, but fall short of E3T, which is the state-of-the-art method for single level ad-hoc single level ad-hoc collaboration performance with humans. While reward is an important metric to optimize, past work (Carroll et al., 2020) has shown that it can obfuscate frustrating behaviors, such as forcing the human to adapt to the agent's strategy, or completing the task effectively but independently, while not truly cooperating with the human. Therefore we must also consider human subjective preferences as equally important in assessing an agents' human-AI cooperation abilities. In Figure 9, we show that across 7 different metrics of human enjoyment, CEC consistently outperforms all baselines, with CEC-FT significantly surpassing E3T ($p < 0.01, t = 3.1233$).

So how is it that CEC obtains such high ratings, but less than SOTA score in the tasks? A closer look at users' qualitative assessments of playing with the different models in Figures 27 gives us a partial explanation for why this may be the case. We see that CEC models were ranked highest by users in terms of their ability to adapt to the participant's behaviors, while also being the least frustrating to work with and most consistent in their actions. Moreover, CEC was rated most enjoyable to cooperate with and best in terms of ability to coordinate. We synthesize the correlation between different users' qualitative ratings to establish a hierarchy of factors most relevant to human's perception of effective cooperation in Overcooked, and show our findings in Figure 10. We report a Cronbach's alpha score of $\approx 0.874$, and use this as a strong signal to validate using qualitative met-

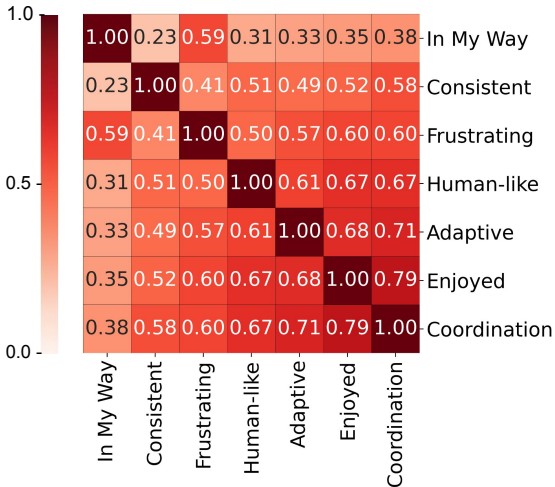

*Figure 10.* Heatmap depicting the Pearson's Correlation Coefficient between users' qualitative rating of different AI collaborators in Overcooked. By clustering based on correlation degree, we can categorize the factors most indicative of what makes a good collaborator. We report a Cronbach's alpha score of $\approx 0.874$.

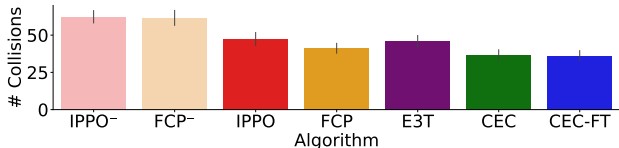

*Figure 11.* Average number of collisions between humans and AI partners on *Counter Circuit* and *Coordination Ring*, with standard error bars shown. CEC achieves the lowest average collision rate.

rics, in addition to reward, as an indicator of CEC's success compared to other baselines.

We also note that CEC's increased proclivity to adapt its behaviors to people can lead to lower rewards if participants are playing a suboptimal strategy. On the other hand, E3T, which has seen the environment during training it is evaluated on, is rated as less adaptive (see Figures 28 and 29), and instead may focus too much on optimizing score. In playing with the agents, we observed that CEC is much better at avoiding collisions and getting out of the player's way, even though this may lead to less reward. Figure 11 quantifies the frequency of collisions between humans and different AI collaborators. The results indicate that, despite achieving lower rewards than the SOTA method E3T, CEC has fewer collisions with humans on average. This aligns with participants' subjective assessments of model behavior (Figure 27). We hypothesize that this **collision-avoidance behavior reflects a general norm learned by CEC**, which enhances performance across a wide range of cooperative tasks. This supports our hypothesis that cross-environment training helps agents learn more generalized cooperative strategies that work not only in a range of environments, but with a range of partners.

## 7. Discussion

We introduce Cross-environment Cooperation, a novel approach to ZSC in multi-agent reinforcement learning. Cross-environment training helps agents learn more generalized cooperative strategies that work not only in a range of environments but with a range of partners. Although trained in self-play, they zero-shot coordinate with new partners *and* in new environments never seen during training. We show that CEC succeeds in cooperating with current state-of-the-art baselines and with human users. These results challenge the prevailing wisdom that self-play is insufficient for ZSC and Ad-hoc Teamplay in cooperative games. Future work will explore composing CEC with other training algorithms. For cooperative AI agents to work effectively with us in our offices and homes, agents must rapidly adapt to a wide range of diverse environments and partners.

## Acknowledgments

We would like to thank the Cooperative AI Foundation, the Foresight Institute, the Amazon+UW Science Gift Hub, the UW Tsukuba NVIDIA Amazon Cross-Pacific AI Initiative and the Sony Resarch Award Program for their generous support of our research. This work was additionally supported by NSF CCF 2212261, NSF IIS 2229881, the Alfred P. Sloan Research Fellowship, and the Schmidt Sciences AI 2050 Fellowship. We are also grateful for this work being supported by a gift from the Chan Zuckerberg Initiative Foundation to establish the Kempner Institute for the Study of Natural and Artificial Intelligence. Lastly, we would like to express our gratitude to our colleagues at the Social Reinforcement Lab and the Computational Minds and Machines Lab for inspiring conversations and insights during the course of this project.

## Impact Statement

This paper presents a novel approach to training agents to collaborate with people it has not seen before. As such, it has the ability to improve the lives of many applications as household robotics, autonomous driving, or language assistants. To provide evidence for the merits of our approach, we conducted studies with human participants. We made sure to follow all ethical practices described under the instructions of our university's Institutional Review Board (IRB), including but not limited to compensating participants at minimum wage and receiving informed consent. As a final point, while the work in this paper described how to build a general cooperative AI, it is not clear or well studied if a similar approach could be used to create a machine capable of harming humans it has not seen before in a wide range of cases. Future works should try to establish whether this is the case and how to mitigate these potential risks.

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

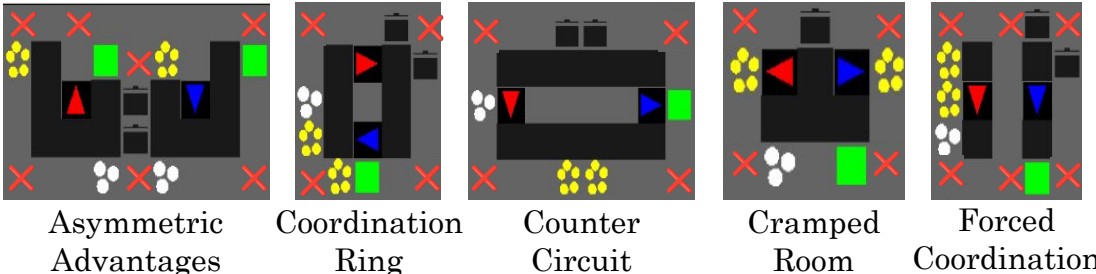

*Figure 12.* Five original Overcooked layouts. The crosses on each of the layouts indicate "unreachable" locations. When sampling layouts in the procedural generation process, we make sure to sample at least one of the plate piles, onion piles, pots, and goals on walls that are reachable

# A. Appendix

### A.1. Procedurally Generated Overcooked

In this section, we provide additional details for how we create many solvable coordination challenges in Overcooked. We first sample which of the five layouts we would like to use as a base predicate. Next, we remove all items and agents from the layout so that we are left with just walls and free spaces. We begin by first sampling a set of plate piles (three white circles in triangle shape in Figure 12), onion piles (three yellow circles in triangle shape in Figure 12), pots (Darker black rectangle with lid embedded within a gray wall in Figure 12), and goals (green squares in Figure 12) to be on a wall not in the regions marked with a red cross in Figure 12. We then sample an additional set of each of the aforementioned items on any of the remaining walls that are not occupied by other objects.

Next, we sample initial agent locations. In the *Coordination Ring, Counter Circuit,* and *Cramped Room* layouts, we can sample both of the agents in any of the available free spaces. However, in *Asymmetric Advantages* and *Forced Coordination* this can be problematic since a divider prevents agents from moving between two halves of the grid. As such, if all task-relevant items are sampled on inaccessible walls, the agents will not be able to complete the tasks. To avoid this issue, for these layouts only we make sure to sample initial agent locations in free spaces on separate halves of the grid. That is, the red agent's initial location will be sampled on a free cell in the left half of the grid and the blue agent's initial location will be sampled on the right half of the grid, or vice versa.

By guaranteeing at least one set of items relevant to the task are reachable and accessible by at least one agent, we can create a new, solvable coordination challenge. We randomly sample whether we will rotate the sampled grid 90 degrees clockwise, and embed the resulting state in a larger 9x9x26 observation space by padding additional cells as just being walls. After making this 9x9x26 grid, we compare it to held-out levels to see whether the goal positions, pot positions, plate pile positions, and onion pile positions are the same. If they are, we resample a new grid. We provide pseudocode for our approach in Algorithm 1.

### A.2. CEC in Partially Observable Environments

We modified the Dual Destination problem to give agents a 3x3 visibility window instead of the fully observable 5x5 window from the results in Figure 3. Then, we trained CEC, FCP, and IPPO using the same architectures as in the fully observable setting and 300 million steps of training. The challenge of breaking handshakes when learning multi-agent policies is even more pronounced in the partially observable case, as agents may form arbitrary conventions to handle high uncertainty about the state of the world. **From our results in Figure 13, we find the same conclusions in the partially observable setting as we did in the fully observable results described in the paper: CEC has high cross-play performance in ZSC with other agents on novel environments (0.74), outperforming population based methods (0.61) and naive self-play (0.03).**

### A.3. CEC in Multi-task Environments

We explored whether CEC would be beneficial for cross-partner and cross-environment generalization when there are multiple solutions to a task. The intuition here is that with multiple possible optimal responses a team of agents could have for collaboration, PBT or self-play methods with sufficient exploration might be able to form robust, object-oriented

---

**Algorithm 1** Solvable Overcooked Coordination Challenge Generation

---

**Input:** Layout set $\mathcal{L} = \{$Coordination Ring, ..., Forced Coordination$\}$, Held-out set of evaluation levels $G_h$
Sample $L_{\text{base}} \sim \mathcal{U}(\mathcal{L})$ {Discrete uniform distribution}
Initialize grid $G \leftarrow$ LoadWalls($L_{\text{base}}$)
$G \leftarrow G \setminus \{$objects $\cup$ agents$\}$
**Phase 1: Mandatory Object Placement**
**for** $\forall o_t \in \mathcal{O} = \{$PlatePile, OnionPile, Pot, Goal$\}$ **do**
    Let $V = \{w \in G_{\text{walls}} \mid w \notin R_{\text{red}}\}$
    $G \leftarrow G \cup \{o_t(\text{rand}(V))\}$
**end for**
**Phase 2: Supplemental Object Placement**
**for** $\forall o_t \in \mathcal{O}$ **do**
    $V_{\text{remaining}} = \{w \in V \mid w \notin G\}$
    $G \leftarrow G \cup \{o_t(\text{rand}(V_{\text{remaining}}))\}$
**end for**
**Agent Positioning**
**if** $L_{\text{base}} \in \{$Asymmetric Advantages, Forced Coordination$\}$ **then**
    $pos_1 \sim \mathcal{U}(\{c \in \text{FreeSpaces}_{\text{left}}\})$
    $pos_2 \sim \mathcal{U}(\{c \in \text{FreeSpaces}_{\text{right}}\})$
**else**
    $pos_1 \sim \mathcal{U}(\text{FreeSpaces})$
    $pos_2 \sim \mathcal{U}(\text{FreeSpaces} \setminus \{pos_1\})$
**end if**
**Post-Processing**
**if** $u \sim \mathcal{U}([0,1]) > 0.5$ **then**
    $G \leftarrow \text{Rotate}(G, 90°)$
**end if**
$G \leftarrow \text{Pad}(G, \text{walls}, 9 \times 9)$
**if** $G \notin G_h$ **then**
    **Return** Valid configuration challenge $G$
**else**
    Repeat generation process
**end if**

---

representations without the need for CEC.

To test this, we extended the Dual Destination environment to have two possible valid solutions to reward agents (Figure 14.) Now, agents are rewarded if they are on opposite green or opposite pink squares. As shown in Figure 15, in the multi-task variant both valid squares remain equidistant from the agents so that there are now 4 strategies which could be rewarded. For the procedural generator, just as in the original Dual Destination problem we randomly shuffle agent and goal locations so that they all lie on unique grid cells. We show **even in the multi-task setting, CEC agents (0.404) outperform PBT methods (0.251) and naive self-play (0.083) when collaborating with novel partners on tasks PBT and naive self-play method swerve trained on. Just as in the single-task setting, PBT (0.005) and naive self-play (0.004) cannot generalize to novel partners on novel environments, whereas CEC can (0.446)**, albeit with a slight performance reduction from the single-task setting in Figure 3 of our paper (CEC=0.931 and 0.966 on fixed and procedurally generated single-task problems respectively). This finding illustrates that additional work is needed to understand the impacts of task complexity and procedural environment generation.

### A.4. Combining Partner and Environment Diversity

We tested the impact of combining partner diversity with environment diversity by training E3T agents under the CEC paradigm. We set the partner policy randomness to 0.5, consistent with human experiments and E3T's original design. Results in the ZSC setting in simulation (Figure 16) show CEC-E3T performs worse than other models on Overcooked's five original layouts (CEC=130.51, CEC-E3T=28.21) but outperforms CEC-Finetune on 100 held-out grids (CEC-FT=41.73,

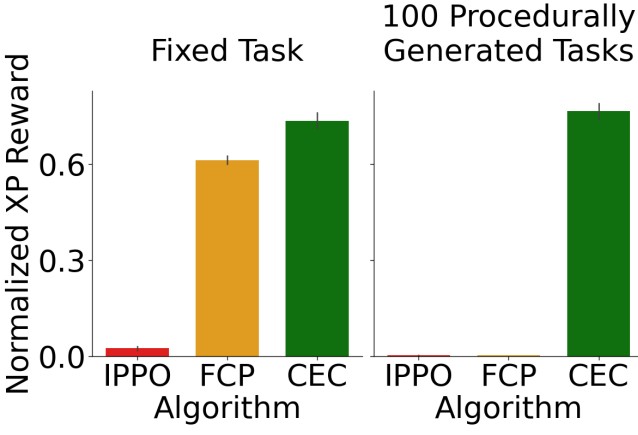

*Figure 13.* In the partially observable setting, CEC ZSC performance on the Dual Destination problem replicates findings from the fully observable case, suggesting it has promise for other games with imperfect information and the need for dynamic conventions, such as Hanabi.

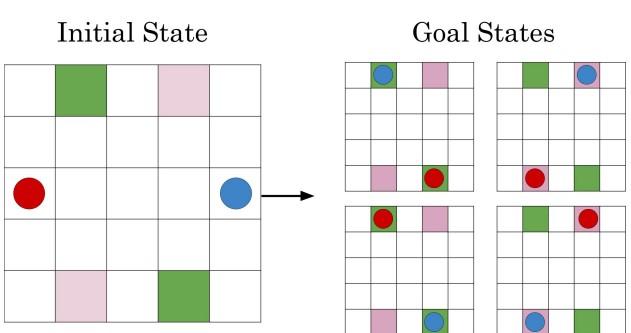

*Figure 14.* Overview of the Multi-Task Dual Destination problem. Agents are rewarded for going to either opposite pink squares or opposite green squares.

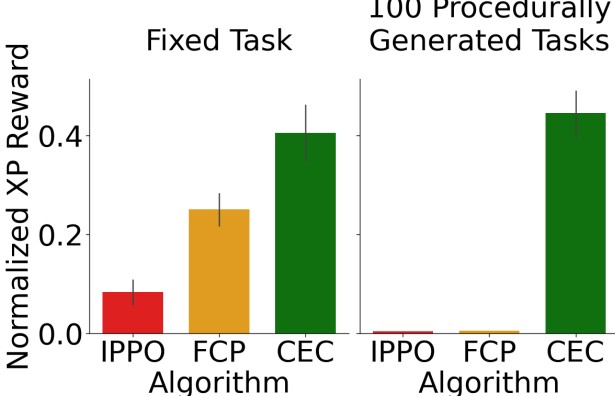

*Figure 15.* Even in this multi-task setting, CEC outperforms population-based methods when tasked with collaborating with novel partners on novel problems. Just as in the single-task setting, naive self-play and PBT methods fail to achieve substantial rewards on environments they have not seen during training, whereas CEC can.

CEC-E3T=58.13). The learning curves (Figures 17 and 18) from a checkpoint 2 billions steps into training reveal that noisy partners introduced additional training noise, which, combined with dynamic environments, likely requires larger networks and longer training times for convergence compared to vanilla CEC.

### A.5. Necessity of Recurrent Networks for CEC

We include recurrent networks with CEC to enable a basic meta-learning algorithm (Wang et al., 2018; Rabinowitz et al., 2018). We tested whether it is possible for CEC to retain reasonable performance without using recurrent policies. In Overcooked and the Dual Destination problem, agents without recurrence failed to converge or adapt effectively. As shown in the learning curves for the Dual Destination problem, CEC with LSTMs successfully converged (Figure 19), while the non-recurrent version couldn't even achieve positive rewards (Figure 20).

### A.6. NiceWebRL for Jax-based Human Experiments

Reinforcement Learning has experienced accelerated progress recently due to the adoption of Jax-based environments that enable tackling single-agent (Bonnet et al., 2024; Nikulin et al., 2024; Matthews et al., 2024) and multi-agent (Rutherford et al., 2023; Lu et al., 2024) problems at millions of steps per second on academic-scale compute. From. To that effect, an open question within the fields of cognitive and computer science is how can these advances in environment design help us progress our understanding of individual and collective agency in humans and machines. For our human experiments we

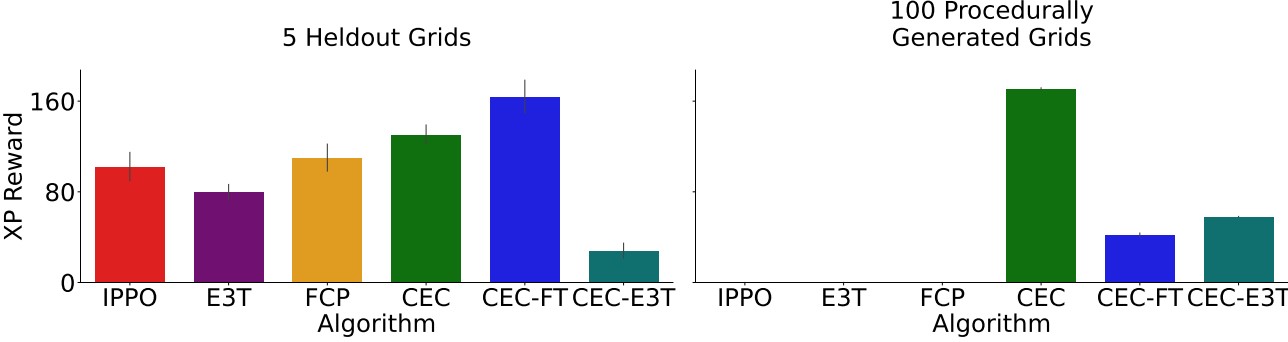

Figure 16. CEC with E3T exhibits lower cross play performance in the ZSC setting than any other method on the original 5 Overcooked layouts, but exhibits better generality to the 100 held out procedurally generated layouts than CEC fine-tuned on one of the original 5

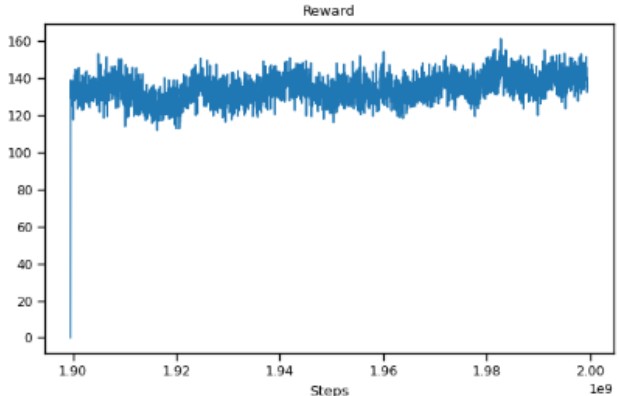

Figure 17. CEC on its own is able to achieve very high rewards during training across novel environments

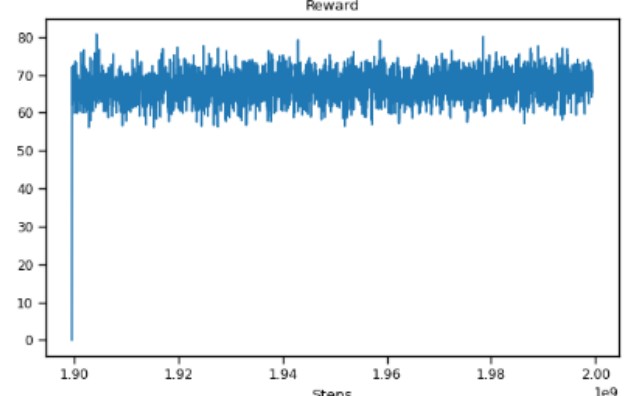

Figure 18. Combining CEC with a partner diversity method (E3T) leads to a training curve which achieves half of the reward as CEC on its own in the same amount of training time

used NiceWebRL (Carvalho, 2025), a unified tool for evaluating single-Human, Human-AI, and Human-Human performance in turn-based or simultaneous-action games.

NiceWebRL is built on top of the NiceGUI nic Python package, and leverages server-side environment parallelization to make a seamless client-side study experience for any single or multi-agent game based in Jax. It precompiles the environment reset and step functions, as well as the code to render the state as a pixel-based image. To ensure minimal latency, NiceWebRL then simulates future possible states and stores an image representing each possible future-world client-side. Then, when the active Human actually takes an action in the game, it renders the corresponding state and simulates forward in time during the few milliseconds before the human can select another action.

Not only does NiceWebRL provide an intuitive way to load in environments and AI models to conduct human experiments on, it also gives researchers an accessible method for collecting and saving participant data thanks to NiceGUI's large variety of survey options. Moreover, it provides a seamless way to deploy experiments on the web from a laptop: even running Human-AI experiments with complex models on a laptop with a few cpu cores, we were able to host several simultaneous user studies across the globe with low latency. The authors of the package originally included just single-agent support, and we extended that package to evaluate human-AI and human-human gameplay.

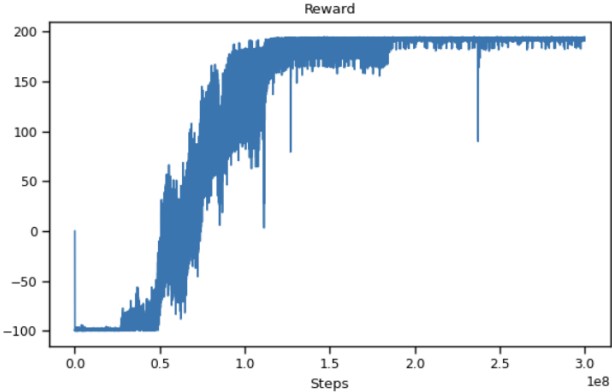

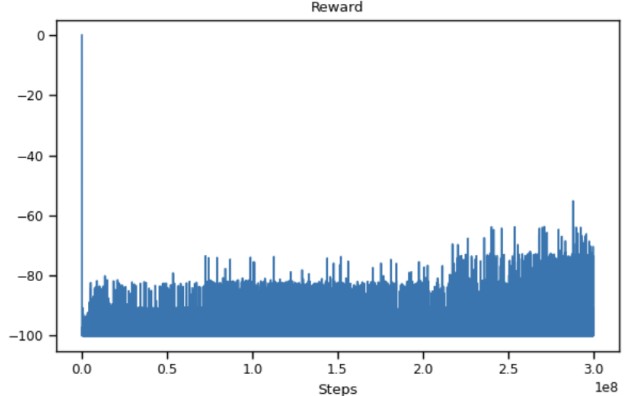

*Figure 19.* In 300 million timesteps, CEC with an LSTM converged to the maximum reward on the Dual Destination problem.

*Figure 20.* In 300 million timesteps, CEC without an LSTM cannot achieve a positive reward on the Dual Destination problem.

## A.7. Agent Training Details

### A.7.1. NETWORK ARCHITECTURE

For all IPPO, FCP, and CEC agents, we use the architecture in Table 1 consisting of three main components: an observation encoder, a recurrent core, and separate actor-critic heads.

| Component | Layer | Details |
|---|---|---|
| Observation Encoder | Conv1 | 2×2 kernel, 64 filters, orth($\sqrt{2}$), ReLU activation |
| | Conv2 | 2×2 kernel, 32 filters, orth($\sqrt{2}$), ReLU activation |
| | FC1 | Fully-connected, 512 units, orth($\sqrt{2}$), ReLU activation |
| | FC2 | Fully-connected, 512 units, orth($\sqrt{2}$), ReLU activation |
| Recurrent Core | LSTM | Feature size: 256, state resets at episode boundaries |
| Actor Head | FC1 | Fully-connected, 256 units, orth(2), ReLU activation |
| | FC2 | Fully-connected, 192 units, orth(2), ReLU activation |
| | FC3 | Fully-connected, 128 units, orth(2), ReLU activation |
| | FC4 | Fully-connected, 64 units, orth(2) [Overcooked only], ReLU activation |
| | Output | Fully-connected, [6 for Overcooked, 5 for Dual Destination Problem] units (logits for discrete actions), orth(0.01) |
| Critic Head | FC1 | Fully-connected, 512 units, orth(2), ReLU activation |
| | FC2 | Fully-connected, 256 units, orth(2), ReLU activation |
| | FC3 | Fully-connected, 192 units, orth(2) [Overcooked only], ReLU activation |
| | FC4 | Fully-connected, 128 units, orth(2) [Overcooked only], ReLU activation |
| | Output | Fully-connected, 1 unit (value prediction), orth(1.0) |

*Table 1.* Agent Architecture for IPPO, FCP, and CEC agents. All layers use orthogonal weight initialization with layer-specific scaling factors (orth(scale)) and zero bias initialization.

For E3T, we use the architecture described in (Yan et al., 2023), with the randomness parameter for the partner policy $= 0.55$

### A.7.2. PPO PARAMETERS

We use the parameters in Table 2 for training all PPO agents:

## A.8. Cross-Algorithm Analysis

Agents trained with each of the different algorithms in the single-task setting (IPPO, FCP, E3T) play each other and the two CEC models on both the five original and 100 procedurally generated held-out Overcooked grids. We show the results in

| Parameter | Value | Description |
|---|---|---|
| LR | $3 \times 10^{-4}$ | Initial learning rate for policy optimization |
| NUM_STEPS | 256 | Number of steps to collect per environment before updating |
| TOTAL_TIMESTEPS | $3 \times 10^{9}$ | Total number of environment steps for training |
| UPDATE_EPOCHS | 4 | Number of epochs to update policy per collected batch |
| NUM_MINIBATCHES | 2 | Number of minibatches to split collected data into |
| GAMMA | 0.99 | Discount factor for future rewards |
| GAE_LAMBDA | 0.95 | Lambda parameter for Generalized Advantage Estimation |
| CLIP_EPS | 0.2 | PPO clipping parameter for policy loss |
| ENT_COEF | 0.005 | Entropy coefficient for encouraging exploration |
| VF_COEF | 1.0 | Value function loss coefficient |
| MAX_GRAD_NORM | 0.5 | Maximum gradient norm for gradient clipping |
| ANNEAL_LR | True | Whether to use learning rate annealing |

*Table 2.* PPO Hyperparameters

Figures 21 and 22. From these heatmaps, we see that CEC can collaborate with agents trained with different algorithms on par with PBT methods, indicating that it is truly learning how to adapt rather than follow a single strategy (Question 1). We use the values in Figures 21 and 22 as the payoff matrices for our Empirical Game Theory analyses in Figure 8.

We additionally include the AI-AI XP performance for agents across each of the original 5 layouts in Figure 23, which shows that CEC or CEC-Finetune perform better than all other models in 4 out of 5 layouts and competitively with other methods in the layout it did not do as well in.

### A.9. Limitations

The learning curves in Figure 7 show that CEC has not yet converged from SP training or plateaued in terms of XP performance on any level. Due to cost and time constraints, we were not able to train for longer than 3 billion steps per model, leaving 2 open questions for future research: 1. What is upper-ceiling on XP performance for IPPO CEC agents, and 2. How can we balance the bias towards adaptation introduced in CEC with greedier policies which maximize reward beyond CEC-Finetune.

Moreover, our human experiments filtered for participants capable of fluently speaking English. This can bias our results, as speaking English typically comes with its own set of cultural practices which impact participants' abilities to play the game and also their assessments of agents' cooperative abilities (Henrich et al., 2010).

### A.10. Qualitative Analysis of Learned Norms

In this section, we provide additional intuition for the success of CEC in comparison to single-task methods. We examine *Counter Circuit* (see Figure 4), a layout with a large amount of strategy diversity. In Figures 24, 25, and 26, we compare the frequency of different locations visited by IPPO and CEC agents in *Counter Circuit*. The prevalence of darker regions for the IPPO agents indicates that they visit certain areas less frequently and tend to follow a fixed route when completing the task. Such strategies can be brittle if people try moving in the opposite direction around the center block, potentially forcing agents into unfamiliar locations where they lack experience acting. In contrast, CECs have a more uniform distribution over the cells they visit. This does not inherently mean CECs are capable of better adaptation, as agents uniformly sampling actions would theoretically also have uniform state coverage. However, we observe that the cells closest to both pots, both onion piles, the plate pile, and the goal location receive the highest concentration of visitations by CEC agents. In contrast, the IPPO agents seem to visit at most one pot. This leads us to infer that CEC agents have developed a richer representation

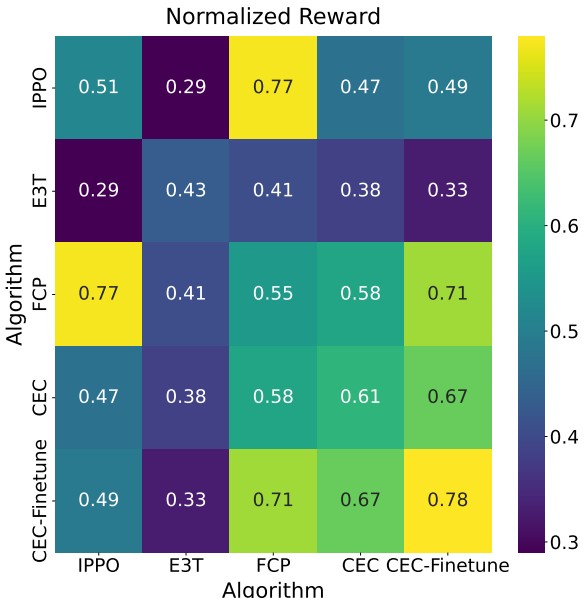 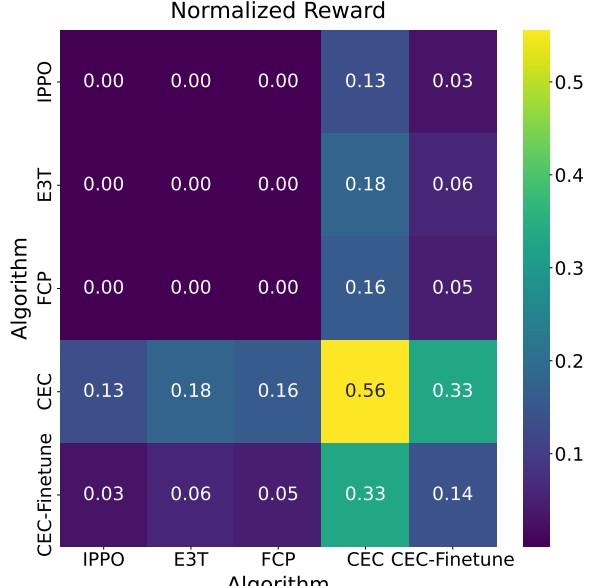

*Figure 21.* Heatmap comparing different algorithms playing each other in the single-task setting, averaged across the original 5 layouts. Brighter yellow regions indicate better XP performance.

*Figure 22.* Heatmap comparing different algorithms playing each other on 100 held-out procedurally generated Overcooked layouts. Brighter yellow regions indicate better XP performance.

of the different subtasks involved in cooking, and may be more capable of adapting to users' actions if forced by the humans to play a different role within the collective plan (i.e. cook items in a different pot, pickup cooked items instead of pass onions, etc.).

### A.11. User Assessments of Partners from Human-AI Experiments

At the end of every episode a participant plays with a different RL agent, we provide them with a survey to assess the agent they just played with. We ask 7 questions that participants respond to with a rating on the Likert scale (Likert, 1932). The questions, and corresponding participant ratings of different models, averaged across the two human-AI experiments we ran (*Counter Circuit* and *Coordination Ring*) are included in Figure 27.

We also include the responses averaged across participants for each individual experiment in Figures 28 and 29.

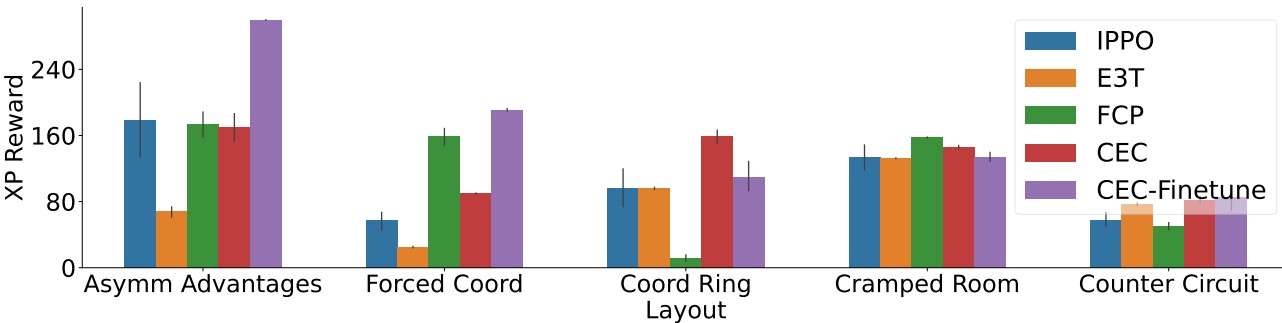

*Figure 23.* Comparison of model performance across each of the original 5 Overcooked layouts, with standard error bars shown. CEC or CEC-Finetune achieve the highest mean reward in 4 out of 5 layouts.

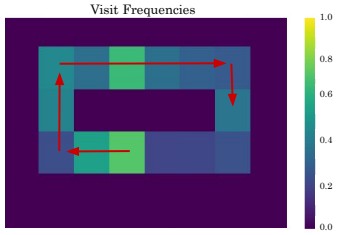

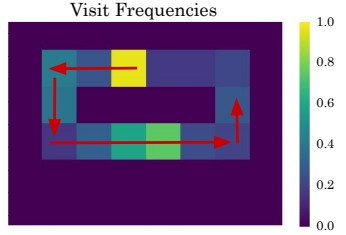

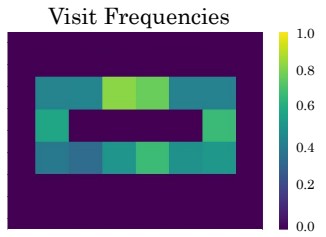

*Figure 24.* IPPO Seed on *Counter Circuit* which has converged upon an optimal strategy of rotating clockwise and the agent at the top doing more of the work. Struggles playing with seed depicted in Figure 25.

*Figure 25.* IPPO Seed on *Counter Circuit* which has converged upon an optimal strategy of rotating counterclockwise and the agent at the bottom doing more of the work. Struggles playing with seed depicted in Figure 24.

*Figure 26.* CEC Seed on *Counter Circuit* which can coordinate with IPPO seeds from Figures 24 and 25. By focusing it's attention on task-relevant objects rather than arbitrary strategies, it's able to effectively cooperate with novel partners on novel problems.

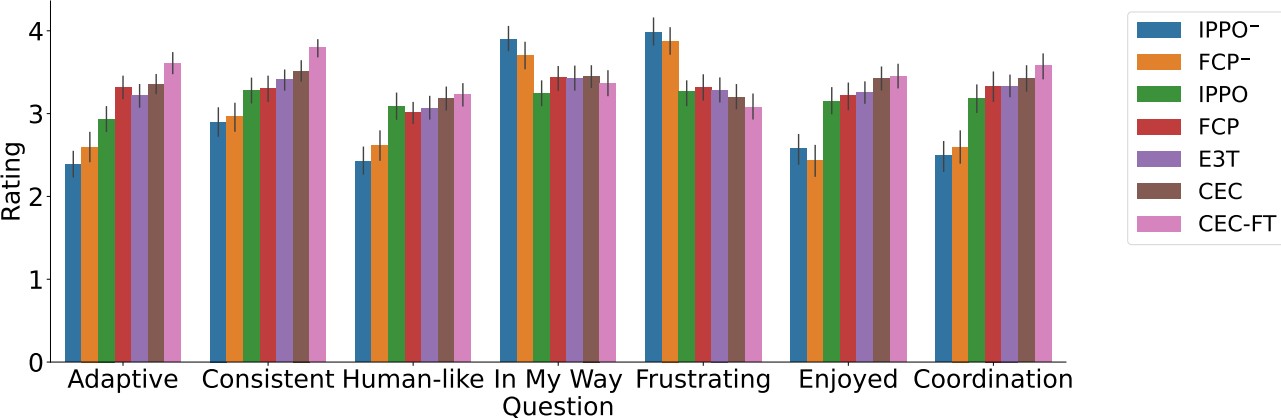

*Figure 27.* Participant assessments of different models across 7 different metrics averaged across *Counter Circuit* and *Coordination Ring*.

## A.12. Human-AI Collaboration Success Per Layout

We plot the success of different models playing with humans in our user study in Figures 30 and 31. We show the results of *Counter Circuit* and *Coordination Ring* respectively.

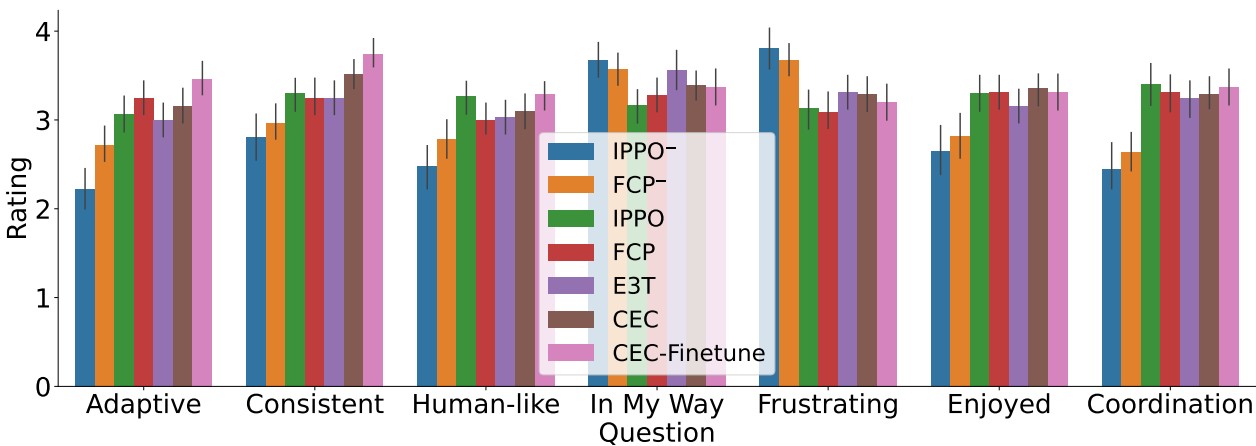

*Figure 28.* Participant assessments of different models across 7 different metrics for the experiment *Counter Circuit*.

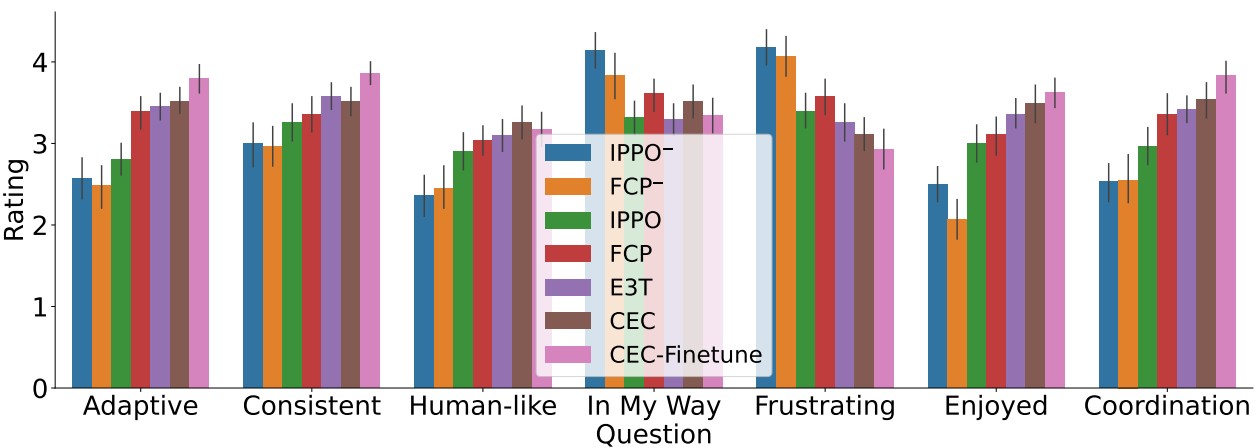

*Figure 29.* Participant assessments of different models across 7 different metrics for the experiment *Coordination Ring*.

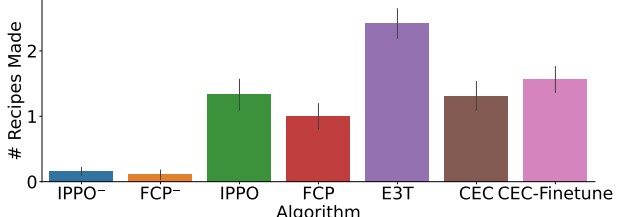

*Figure 30.* Success rates of different algorithms playing *Counter Circuit* with humans.

*Figure 31.* Success rates of different algorithms playing *Coordination Ring* with humans.

