# OpenReview forum: "Cross-environment Cooperation Enables Zero-shot Multi-agent Coordination"
_ICML.cc/2025/Conference — ICML 2025 oral_

### Official Review · Reviewer_H4DQ · 2025-02-17

**Overall Recommendation:** 3

**Summary:**

This paper proposes to train agents in self-play on a large-distribution of environments to enhance the agents' coordination ability with unseen teammates in unseen environments. Experiments on a toy grid-world game and Overcooked demonstrate the effectiveness of the proposed method.

**Claims And Evidence:**

Yes.

**Essential References Not Discussed:**

No.

**Experimental Designs Or Analyses:**

Yes. I checked Section 5. Experiments and Section 6. Results.

**Methods And Evaluation Criteria:**

Yes.

**Other Comments Or Suggestions:**

None.

**Other Strengths And Weaknesses:**

### Strengths
1. **Clarity and Simplicity:** The paper is well-written and easy to follow. The proposed method, CEC, is presented in a simple and straightforward manner, making it accessible to readers.
2. **Empirical Validation:** The paper provides extensive experimental results that demonstrate the effectiveness of CEC in enhancing coordination generalization.
3. **Interesting Insight:** The observation that training on multiple environments (initial states) improves coordination with unseen teammates is intriguing and potentially impactful for zero-shot coordination (ZSC) tasks.

### Weaknesses
1. **Lack of Analysis:** The fundamental reason why training on multiple environments enhances coordination generalization remains unclear. The paper would benefit from a more rigorous theoretical analysis or analytical experiments to explain this phenomenon. Without such analysis, it is difficult to assess whether CEC can generalize to more complex tasks.
2. **Limited Novelty:** The proposed method appears to be a direct application of multi-task RL to ZSC, which lacks significant novelty. Additionally, key methodological details, such as the task sampling strategy, are not clearly explained.
3. **Narrow Experimental Scope:** CEC is evaluated only on two simple discrete grid-world environments, where enumerating initial states is straightforward. In more complex scenarios, the cost of creating diverse environments may outweigh the benefits of training with unseen teammates, potentially limiting the practical applicability of CEC.

**Questions For Authors:**

See Weaknesses.

**Relation To Broader Scientific Literature:**

The key contributions extend and refine ideas from MARL, cooperative AI, game theory, robustness in machine learning, emergent communication, theory of mind, and open-ended learning. By addressing the challenge of generalization to unseen partners and environments, this paper advances these fields and provides a framework for developing more adaptable and collaborative AI systems.

**Theoretical Claims:**

There is no formulated theoretical claims in this paper.

---

> ### Author Rebuttal · Authors · 2025-04-01
>
> Thank you for your time and effort in reviewing our paper, and for recognizing its strengths in clarity, empirical validation, and the intriguing insights it provides.
>
> # New experiments
>
> Based on your suggestions, we’ve added the following results to broaden the experimental scope to more complex scenarios, and improve the analytical strength of our work:
>
> - **[CEC in Partially Observable Environments](https://rb.gy/iwg3wx).** Testing CEC in a partially-observed version of the Dual Destination problem yielded similar results: CEC achieved high cross-play performance on novel environments (0.74), outperforming population-based methods (0.61) and naive self-play (0.03).
> - **Combining Task and Partner Diversity.** Following reviewer mZRY and 11py’s suggestions, we tested combining [CEC with E3T](https://rb.gy/g015me). While it performed poorly on the original Overcooked layouts (CEC=130.51, CEC-E3T=28.21), it outperformed CEC-Finetune on held-out grids (CEC-FT=41.73, CEC-E3T=58.13). We hypothesize the E3T partner noise will require larger networks and more training time when combined with CEC’s diverse environments.
> - **Analyzing the effect of RNNs in CEC.** Following reviewer SmWr’s suggestions, we provide new experiments ablating the use of an RNN for CEC, where the RNN is used to provide a simple meta-learning algorithm (Wang et al, 2018; Rabinowitz et al, 2018). In Overcooked and the Dual Destination problem, agents without recurrence failed to converge or adapt effectively. As shown in the [learning curves for the Dual Destination problem](https://rb.gy/nti7xf), CEC with LSTMs successfully converged, while the non-recurrent version could not even achieve positive rewards.
>
> # Overcooked benchmark
>
> Following many prior papers published at top conferences like ICML (Li et al (2023); Mahlau et al. (2024)), NeurIPS (Carroll et al. (2020), Strouse et al. (2022), Yan et al. (2023),  Sarkar et al. (2023), Myers et al. (2024), Liang et al. (2024)) and ICLR (Yu et al. (2023), Gessler et al. (2025)), we focus on the Overcooked benchmark as a challenging human-AI cooperation task. *A core strength of Overcooked is that it enables real-time evaluation with actual human players, and shows that even for the simplest layouts coordinating with heterogenous and unpredictable human players requires a level of robustness that even state-of-the-art AI algorithms typically do not achieve.* **Since your review did not mention our human-study results** (Figures 9-11), **we would like to draw your attention to the fact that CEC achieves performance equivalent to state-of-the-art techniques in real human evaluations *without using population-based training,*** a surprising finding that contradicts much of the human-AI coordination literature.
>
> # Novelty
>
> While multi-task RL is well-established (Tobin et al., 2017), **our work is, to our knowledge, one of the first to show that environment diversity can outperform partner diversity for human-AI coordination, challenging the dominance of Population-Based Training (PBT) in ZSC** (Vinyals et al., 2019; Carroll et al., 2020; Strouse et al., 2022; Zhao et al., 2022; Sarkar et al., 2023; Liang et al., 2024). To support this claim, we reference Reviewer 11py’s comment, “I really like the idea of training the learning agent across variations of environments, which **has not been investigated much, if at all, in the literature. I think the authors could also go as far as claiming cross-environment cross-play evaluation as a novel evaluation setup, which would serve as another contribution of the paper.”**
>
> # Methodological details
>
> Our task sampling strategy is described in Section 4 and Appendix A.1. Figure 12 details how we determine which items to sample to create solvable environments, and how this leads to   procedural generation of over $1.16 * 10^{17}$ diverse, solvable coordination challenges by sampling wall structures and randomizing features like goals, plates, pots, and onions.
>
> Our paper provides several **insights into how and why training in multiple environments enhances coordination with many partners:**
>
> - **Real Human Experiments:** Quantitative and Qualitative assessments (Section 6, Figures 9-11, 16-21) indicate agents learn to "adapt to partners in service of completing the task."
> - **Heat maps** (Figures 16-18) visualize the frequency of states covered by naive IPPO agents and CEC. CEC agents are able to adapt to novel partner strategies better by covering a greater distribution of states in self-play.
> - **Empirical game-theoretic analysis** (Section 5) demonstrates CEC agents forming more robust equilibria with novel partners.
> - **Learning curves** (Figure 7) illustrate how CEC agents improve differently on held-out tasks based on optimal strategy diversity.
>
> Thank you again for your feedback, which will help us improve the final version of our paper.

---

> > ### Comment · Reviewer_H4DQ · 2025-04-03
> >
> > My concerns are largely addressed. I raise my score from 1 to 3.

---

> > > ### Author Response · Authors · 2025-04-08
> > >
> > > Thank you for taking the time to engage with our paper and updating your score! Please let us know if there are any additional questions or concerns we can address. We wanted to highlight an additional experiment we ran during this second rebuttal period that might be of interest.
> > >
> > > # CEC in multi-task environments
> > >
> > > Following reviewer 11py’s suggestions, we explored whether CEC would be beneficial for cross-partner and cross-environment generalization when there are multiple solutions to a task. The intuition here is that with multiple possible optimal responses a team of agents could have for collaboration, PBT or self-play methods with sufficient exploration might be able to form robust, object-oriented representations without the need for CEC.
> > >
> > > To test this, [we extended the Dual Destination environment](https://rb.gy/4f9hmh) to have two possible valid solutions to reward agents. Now, agents are rewarded if they are on opposite green or opposite pink squares. As shown in the attached Figure, in the single-task variant, both valid squares remain equidistant from the agents so that there are now 4 strategies which could be rewarded. For the procedural generator, just as in the original Dual Destination problem we randomly shuffle agent and goal locations so that they all lie on unique grid cells.
> > >
> > > We show, even **in the multi-task setting, CEC agents (0.404) outperform PBT methods (0.251) and naive self-play (0.083) when collaborating with novel partners on tasks PBT and naive self-play method swerve trained on. Just as in the single-task setting, PBT (0.005) and naive self-play (0.004) cannot generalize to novel partners on novel environments, whereas CEC can (0.446)**, albeit with a slight performance reduction from the single-task setting in Figure 3 of our paper (CEC=0.931 and 0.966 on fixed and procedurally generated single-task problems respectively). As the reviewer insightfully hinted at, this finding illustrates that additional work is needed to understand the impacts of task complexity and procedural environment generation, and we will use these results to create a more nuanced characterization of the generalizability of our work.

---

### Official Review · Reviewer_SmWr · 2025-03-11

**Overall Recommendation:** 5

**Summary:**

This work presents cross-environment coordination as an alternative to population based training for enabling smooth coordination with unseen partners. They find that (pre-)training on a diverse set of environment configurations on Overcooked with a single learning partner enables agents to work in new environments with new partners. A human-AI user study demonstrates the effectiveness of their training approach for human-AI coordination.

## Updates after Rebuttal

The authors have clarified the ZSC vs AHT distinction I emphasized in my original review, and added new results to demonstrate that CEC requires an RNN to train. The experimental results in this paper are very interesting, so I strongly recommend this paper be accepted.

**Claims And Evidence:**

I detail specific claims that I find problematic in later sections of the review.

**Essential References Not Discussed:**

The related works section is comprehensive (assuming the setting is ad-hoc coordination and not ZSC)

**Experimental Designs Or Analyses:**

The experiments and analyses are valid (with the exception of the issue mentioned earlier in the evaluation criteria)

**Methods And Evaluation Criteria:**

- This may just be a misunderstanding, but it is incorrect to claim that training multiple agents using the “same algorithm” suffices for XP evaluation. For instance, the first cited work in the XP Evaluation section (Strouse et al., 2022) measures performance by explicitly testing against a held-out set of agents (H_proxy, diverse SP, and random agents) and does not test against the same algorithm being trained.
     - Cross-seed XP on the same algorithm is only valid in the ZSC setting, not ad-hoc teamplay (see my comments in “other comments or suggestions” for the distinction).

**Other Comments Or Suggestions:**

- I think the references to ZSC should be replaced with ad-hoc coordination, based on conventions from the MARL literature. Although similar, ZSC refers to the setting where we assume partners follow the same algorithm and attempt to maximize cross-play across initializations, while ad-hoc coordination refers to the ability to adapt to new partners who may not share the same learning algorithm. In the context of human-AI coordination, it seems like this paper cares more about the latter. Please refer to “‘Other-Play’ for Zero-Shot Coordination” in ICML 2020 for more information.
- Line 161 column 1: “we how training” typo
- A should not be duplicated in the Markov Game tuple definition. The horizon should also be defined in the Markov Game tuple, especially since it is used in the score definition (though I would advise against using T for both transitions and horizon).

**Other Strengths And Weaknesses:**

Strengths:
- This is a very interesting work, demonstrating strong results for cross-environment coordination helping for ad-hoc coordination.
- The human-AI experiments are principled and demonstrate strong transfer to human partners.
- The empirical game-theoretic analysis is a very creative and interesting way to demonstrate cross-algorithm performance.

Weaknesses:
- The most critical weakness is the confounding of ZSC and ad-hoc coordination. It seems like this paper motivates itself under the (harder) ad-hoc coordination paradigm but evaluates itself under the (easier) ZSC paradigm.
    - I would typically request using held-out partners for these results (i.e. the human BC model used for evaluation in Overcooked on earlier works), but the human user study is sufficient. I’d instead like to see the ZSC vs ad-hoc coordination point clarified throughout the text.
- This work only studies fully observable, simultaneous action settings, but coordination challenges occur in partially observable settings (like Hanabi) so we cannot generalize the results of this paper to the broader ZSC community.

**Questions For Authors:**

- My interpretation of the key results is that the pair of agents trained via CEC do not learn consistent “conventions” across environments, so they “give up” and don’t learn any conventions. Are there experiments to indicate that CEC models do have some underlying conventions consistently across environments?
    - In particular, I am wondering about settings where forming conventions is more strictly necessary (i.e. settings with partial observability), and whether CEC helps create consistent conventions.
- Given that Overcooked is fully observable, how important is the recurrent core for performance and generalization? This seems to be a key difference between this work and (Yan et al., 2023), and this may be hurting the human-AI performance of your methods since it is easier to move off distribution at test time with recurrent inputs.

**Relation To Broader Scientific Literature:**

The key contribution is applying domain randomization to the multi-agent setting for the purposes of ad-hoc coordination, which is novel to my understanding.

**Theoretical Claims:**

N/A

---

> ### Author Rebuttal · Authors · 2025-04-01
>
> We would like to thank the reviewer for their thoughtful and constructive feedback on our paper, and address your comments below:
>
> # ZSC vs Ad-hoc Teamplay
>
> We sincerely thank you for clarifying the distinction between these two evaluation settings. From our understanding, our use of the empirical game theoretic analysis to compare how agents trained with different algorithms cooperate with each other, as well as our human study, both fall within the ad-hoc teamplay setting. However, we acknowledge  that some of our experiments (particularly Figures 3, 5, and 7) address the ZSC setting, where we evaluate performance against the same algorithm with different random seeds. We will follow your suggestions to clarify our references and experimental analysis to distinguish between the two, and make a more nuanced characterization of the conclusions we can draw. We will also make sure to reference "Other-Play.”
>
> # CEC in Partially Observable Environments
>
> Following your suggestions, we tested [CEC with partial observability](https://rb.gy/iwg3wx), by modifying the Dual Destination problem. We trained CEC, FCP, and IPPO using the same architectures and 300 million steps of training. The challenge of breaking handshakes when learning multi-agent policies is even more pronounced in the partially observable case, as agents may form arbitrary conventions to handle high uncertainty about the state of the world. **From our results below, we find the same conclusions in the partially observable setting as we did in the fully observable results described in the paper: CEC has high cross-play performance in ZSC with other agents on novel environments (0.74), outperforming population based methods (0.61) and naive self-play (0.03).** We believe this finding strengthens our paper, and thank the reviewer for providing this suggestion.
>
> # New experiments with no RNN
>
> We include recurrent networks with CEC to enable a basic meta-learning algorithm  (Wang et al, 2018; Rabinowitz et al, 2018). Following your suggestion, we tested whether it is possible for CEC to retain reasonable performance without using recurrent policies. In Overcooked and the Dual Destination problem, agents without recurrence failed to converge or adapt effectively. As shown in the [learning curves for the Dual Destination problem](https://rb.gy/nti7xf), CEC with LSTMs successfully converged, while the non-recurrent version couldn't even achieve positive rewards.
>
> # Combining Task and Partner Diversity
>
> Following reviewer mZRY and 11py’s suggestion, we tested [combining partner diversity algorithms with environment diversity](https://rb.gy/g015me) using E3T under the CEC paradigm. Results showed CEC-E3T performed worse than other models on Overcooked’s five original layouts (CEC=130.51, CEC-E3T=28.21) but outperformed CEC-Finetune on 100 held-out grids (CEC-FT=41.73, CEC-E3T=58.13). The linked learning curves reveal that noisy partners combined with dynamic environments introduced additional noise, suggesting larger networks and longer training times for convergence may be needed compared to vanilla CEC.
>
> # Learned Conventions in CEC
>
> You raised an interesting point about learned conventions from CEC. As noted in our qualitative analysis, conventions like "move out of the way" emerged from CEC agents trained across diverse environments. Unlike fixed strategies (e.g., "red agent cooks onions while blue delivers"), this adaptability reflects a general form of learned convention that enhances transferability and generalization to novel scenarios.
>
> We appreciate your attention to detail in pointing out the typos and suggestions for improving our Markov Game tuple definition. We will address these in our revision.
>
> We thank you for recognizing the strengths in our work, particularly the demonstration of cross-environment coordination's benefits for ad-hoc coordination, the principled human-AI experiments showing strong transfer to human partners, and the creative use of empirical game-theoretic analysis to demonstrate cross-algorithm performance. We thank the reviewer again for their valuable feedback, which will help us improve the clarity and rigor of our paper.
>
> Ref:
> Rabinowitz, N., Perbet, F., Song, F., Zhang, C., Eslami, S. A., & Botvinick, M. (2018, July). Machine theory of mind. In International conference on machine learning (pp. 4218-4227). PMLR.
> Wang JX, Kurth-Nelson Z, Kumaran D, Tirumala D, Soyer H, Leibo JZ, Hassabis D, Botvinick M. Prefrontal cortex as a meta-reinforcement learning system. Nat Neurosci. 2018 Jun;21(6):860-868. doi: 10.1038/s41593-018-0147-8. Epub 2018 May 14. PMID: 29760527.

---

> > ### Comment · Reviewer_SmWr · 2025-04-01
> >
> > Thank you for the new experiments and for updating the text to clarify the ZSC vs AHT distinction.
> >
> > As a follow-up, it is *extremely* surprising to me that the LSTM is necessary for CEC given that the settings are fully observable and there is only one partner. Even complex, multi-task, fully-observable, single-agent environments don't need recurrence ("Kinetix: Investigating the Training of General Agents through Open-Ended Physics-Based Control Tasks" in ICLR 2025 comes to mind), so there may be something special about MARL that necessitates the "meta-learning" capabilities enabled by recurrence.

---

> > > ### Author Response · Authors · 2025-04-08
> > >
> > > Thank you for your engagement with our rebuttal and for updating your review! We believe the reason the RNN is necessary in the MARL case is that additional non-stationarity beyond the environment changing is introduced through a partner causally impacting subsequent observations. With many equally valuable strategies a cooperative partner may adopt (such as move clockwise vs counterclockwise), conditioning on the history of past states is needed to overcome difficulties in predicting the future that only focusing on the current state faces. This does not necessarily mean using an RNN: as you pointed out in your initial review, E3T did not use any recurrence in their architecture. However, they still conditioned on the past 5 states to form a character embedding that was used to model another agent’s actions, then conditioned on the current state and the partner character embedding to perform well in a highly non-stationary learning environment, effectively conditioning on the history of states.
> > >
> > > Here, we see the meta-learning problem as using the first few steps of the episode to adapt to a new partner in a new environment, which is only possible for models that can condition on the episode history to revise their policy.

---

### Official Review · Reviewer_11py · 2025-03-11

**Overall Recommendation:** 5

**Summary:**

This paper studies a novel multi-agent training paradigm, Cross-Environment Cooperation (CEC), where the learning agent learns to work with a single partner across different variations of the environment. This is in contrast with prior work in the literature that focuses on training an agent that can adapt to unseen partners/strategies under a fixed environment. Despite not training with diverse partners, the CEC agents outperform state-of-the-art baselines under fixed and procedurally generated layouts. Additionally, the paper utilizes a newly developed Jax based procedural 2-player Overcooked environment for efficient training.

**Claims And Evidence:**

All the claims are well supported by convincing evidence.

**Essential References Not Discussed:**

-

**Experimental Designs Or Analyses:**

All experiments are well designed and provide good intuition for the reader. The analyses are sound and based on best statistical practices.

**Methods And Evaluation Criteria:**

The proposed method is technically sound and has strong implication to the MARL literature. The evaluation protocol is clear and reasonable. All methods use the same or similar computation budget for a fair comparison.

**Other Comments Or Suggestions:**

Suggestions
- I really like the idea of training the learning agent across variation of environments, which has not been investigated much, if at all, in the literature. I think the authors could also go as far as claiming cross-environment cross-play evaluation as a novel evaluation setup, which would serve as another contribution of the paper. In its current form, the paper reads like CEC is purely a novel training paradigm.
- In Section 4, since there are only two possible strategies, it would be beneficial to include a fixed task oracle baseline. The oracle cooperator would be trained against the two strategies. This would make it clear to the reader how much environment diversity helps partner generalization relative to this oracle (which is trained with "maximally diverse partners").
- Is it possible to show some combination of a partner diversification method (e.g., FCP) and CEC? Analyzing if the diversity generated from the two sources are additive or redundant would provide a very valuable insight
-  Since the Overcooked environment used in the paper has only one recipe, It is quite straightforward to see why varying environment helps: the agents learn to do the recipe in different orders/combinations under different environment variants. I wonder if CEC would work well in environments with multiple solutions (e.g., multiple recipes in Ovecooked [1,2,3]).

typos
- line 161: "In contrast, we how training ..."


[1] Wu, Sarah A., et al. "Too many cooks: Bayesian inference for coordinating multi‐agent collaboration." Topics in Cognitive Science 13.2 (2021): 414-432.
[2] Charakorn et al. "Generating diverse cooperative agents by learning incompatible policies." ICLR. 2023.
[3] Yu, Chao, et al. "Learning Zero-Shot Cooperation with Humans, Assuming Humans Are Biased." ICLR. 2023.

**Other Strengths And Weaknesses:**

Strengths
- The paper is well written and easy to follow.
- The core idea of the paper is novel, simple and very effective.
- The toy example provides good intuition.
- The experiments are well designed and thorough.

Weaknesses
- The environments used in this work are limited and relatively simple.

**Questions For Authors:**

- There are two axes of generalization considered in this paper. It shows that environment diversity helps partner generalization. I wonder if the opposite is true: Does training with diverse partners help environment generalization?
- Why E3T has lower evaluation XP performance than IPPO in Fig. 6 (left)?
- How come E3T gets the lowest XP reward while perform best with humans?
- Will the code be open source?

**Relation To Broader Scientific Literature:**

The findings in this paper will be impactful in the field of ad-hoc teamwork and multi-agent reinforcement learning in general. The idea of positive transfer between the two axes of generalization (environment and partner) is an intriguing finding. This paper serves as a good first step towards achieving jointly environment and partner generalization.

**Theoretical Claims:**

N/A

---

> ### Author Rebuttal · Authors · 2025-04-01
>
> Thank you for your interest in our work and recognizing the novelty of our training and evaluation approaches. We are glad you found our writing to be clear and our experiment section to be thorough. Your idea for framing cross-environment cross-partner evaluations as a novel contribution is one we will take on board in our revision of the paper, and we feel this will make our paper even stronger. We would now like to address some of their questions below:
> # Combining Task and Partner Diversity
> Following your suggestion, we tested the impact of [combining partner diversity with environment diversity](https://rb.gy/g015me) using E3T under the CEC paradigm. We set the partner policy randomness to 0.5, consistent with human experiments and E3T’s original design. Results in simulation showed CEC-E3T performed worse than other models on Overcooked’s five original layouts (CEC=130.51, CEC-E3T=28.21) but outperformed CEC-Finetune on 100 held-out grids (CEC-FT=41.73, CEC-E3T=58.13). The attached learning curves reveal that noisy partners introduced additional training noise, which, combined with dynamic environments, likely requires larger networks and longer training times for convergence compared to vanilla CEC. While the vanilla E3T struggled in simulation, it excelled in human trials, outperforming all other models in terms of reward.
> # Fixed Task Oracle
> In Figure 3 of the paper, we report the score in Dual Destination normalized by the theoretical maximum possible score which would be achieved if both agents spawned on the correct goals in the first timestep. Figure 3 plots these normalized cross-play rewards for CEC, FCP, and IPPO. To obtain the **oracle score that you suggested on the Fixed Task,** we calculate that **an oracle agent** that perfectly responds to either optimal strategy would require 3 steps of receiving a -1 step cost to move to the target location before receiving a (3 positive - 1 step cost) reward for 97 steps, equating to (2*97 - 3) = 191 / 200 = **0.955 normalized reward**. **CEC scored 0.931 normalized reward** with a standard error of 0.013, indicating it underperforms the oracle’s cross play performance by about 2.5%.
> # CEC in Partially Observable Environments
> We tested [CEC with partial observability](https://rb.gy/iwg3wx), by modifying the Dual Destination problem. We trained CEC, FCP, and IPPO using the same architectures and 300 million steps of training. The challenge of breaking handshakes when learning multi-agent policies is even more pronounced in the partially observable case, as agents may form arbitrary conventions to handle high uncertainty about the state of the world. **From our results below, we find the same conclusions in the partially observable setting as we did in the fully observable results described in the paper: CEC has high cross-play performance in ZSC with other agents on novel environments (0.74), outperforming population based methods (0.61) and naive self-play (0.03).** We believe this finding strengthens our paper, and thank the reviewer for providing this suggestion.
> # New experiments with no RNN
> Following reviewer SmWr’s suggestions, we provide new experiments ablating the use of an RNN for CEC, where the RNN is used to provide a simple meta-learning algorithm (Wang et al, 2018; Rabinowitz et al, 2018). We find that in Overcooked and the Dual Destination problem, agents without recurrence failed to converge or adapt effectively. As shown in the [learning curves for the Dual Destination problem](https://rb.gy/nti7xf), CEC with LSTMs successfully converged, while the non-recurrent version could not achieve positive rewards.
> # E3T performance difference in cross-play performance in simulation vs with humans
> During training, E3T uses a partner policy defined as π_p = (1-ε) * learned_policy + ε * uniform_policy, maintaining entropy to induce greater strategy coverage without requiring diverse co-players like population-based methods. Yan et al. (2023) found ε = 0.5 most effective for collaborating with humans, which we used for our baseline. For AI zero-shot coordination, ε = 0.3 performed best, while ε = 0.0 excelled in low-exploration layouts like Forced Coordination.
>
> **The code will be open sourced, from the environment code to training scripts to the human evaluation interface, which supports running experiments on arbitrary Jax-based reinforcement learning environments.**

---

> > ### Comment · Reviewer_11py · 2025-04-02
> >
> > I appreciate the detailed response from the authors. I find the "Combining Task and Partner Diversity" and "Fixed Task Oracle" experiments very informative. I strongly suggest the authors put these results in the paper. I also do really appreciate that the source code will be fully open source.
> >
> > There are still some questions and concerns not addressed. I keep my score as is.

---

> > > ### Author Response · Authors · 2025-04-08
> > >
> > > Thank you for engaging in the rebuttal process. We will follow your suggestions and include the new results from our previous response in the paper. We would like to address some of their questions and concerns from your initial review below:
> > >
> > > # CEC in multi-task environments
> > >
> > > Following your suggestions, we explored whether CEC would be beneficial for cross-partner and cross-environment generalization when there are multiple solutions to a task. The intuition here is that with multiple possible optimal responses a team of agents could have for collaboration, PBT or self-play methods with sufficient exploration might be able to form robust, object-oriented representations without the need for CEC.
> > >
> > > To test this, [we extended the Dual Destination environment](https://rb.gy/4f9hmh) to have two possible valid solutions to reward agents. Now, agents are rewarded if they are on opposite green or opposite pink squares. As shown in the attached Figure, in the single-task variant, both valid squares remain equidistant from the agents so that there are now 4 strategies which could be rewarded. For the procedural generator, just as in the original Dual Destination problem we randomly shuffle agent and goal locations so that they all lie on unique grid cells.
> > > 	We show **even in the multi-task setting, CEC agents (0.404) outperform PBT methods (0.251) and naive self-play (0.083) when collaborating with novel partners on tasks PBT and naive self-play method swerve trained on. Just as in the single-task setting, PBT (0.005) and naive self-play (0.004) cannot generalize to novel partners on novel environments, whereas CEC can (0.446)**, albeit with a slight performance reduction from the single-task setting in Figure 3 of our paper (CEC=0.931 and 0.966 on fixed and procedurally generated single-task problems respectively). As the reviewer insightfully hinted at, this finding illustrates that additional work is needed to understand the impacts of task complexity and procedural environment generation, and we will use these results to create a more nuanced characterization of the generalizability of our work.
> > >
> > > # Does partner diversity improve environment generalization
> > >
> > > As discussed in Section 6 question 2 of our paper, and demonstrated in Figures 3, 6, 8, 9, and 14, we found that **partner diversity on its own does not improve environment generalization.** This is likely due to the fact that on a single task, irrespective of the number of diverse partners an ego cooperator is exposed to, it will struggle to form robust representations of the optimal policy in a way that supports state generalization, since it can associate its learned behaviors with brittle functions such as “move in pattern A then B” rather than in object-oriented or task-centric ways. There is much additional work needed on how to combine partner and environment diversity to realize the full benefits of both.

---

### Official Review · Reviewer_mZRU · 2025-03-11

**Overall Recommendation:** 3

**Summary:**

This paper proposes Cross-Environment Generalization (CEC) as a way of improving agents’ generalization to unseen agents (the ad hoc teamwork problem) and unseen environments.
The proposed method consists of a procedurally generator that varies Overcooked initial states, over which an IPPO team learns via self-play.
The method is evaluated on Overcooked, and demonstrates improved task generalization compared to FCP and E3T, and generalization to human teammates that improves over FCP, but is equivalent to E3T.

**Claims And Evidence:**

- **Key Claim**: training agents in across procedurally generated, randomized environments, improves agents’ ability to generalize to new environments and new cooperation partners.

    - This is the core hypothesis/claim of the paper, but I think it is overstated. The experiments only analyze CEC’s performance on Overcooked, and do not consider any other environments. The efficacy of environment randomization in inducing diverse partner strategies might be specific to Overcooked, so I think the authors should weaken their claims.

- **Secondary Claim**: The authors claim that their procedural generation enviornment generation method is superior to randomly generating Overcooked environments, due to the problem of generating unsolveable environments.

    - However, there is no empirical evidence provided that shows that random level generation is problematic. Can the authors empirically test what percentage of randomly generated environments would be unsolvable in Overcooked?

- **Secondary Claim**: training on large set of procedurally generated environments is easier/results in a more computationally efficient algorithm than training a population of agents on a small set of environment configurations:

    - CEC is trained for 3 billion steps on Overcooked. Isn’t this a much larger training duration than conventional teammate generation approaches such as LIPO and CoMeDi (generally around a couple million timesteps, with a population size < 10) on Overcooked?

**Essential References Not Discussed:**

MADRID (Samvelyan et al. 24) - see above for citation.

**Experimental Designs Or Analyses:**

Overall, the empirical design/analysis is strong, other than the issues I described in Claims/Evidence.

**Methods And Evaluation Criteria:**

The proposed method, CEC is framed as an algorithm for training ad hoc agents, but environment diversity seems somewhat orthogonal from teammate diversity, even if environment diversity can induce teammate diversity. As such, it seems unfair to evaluate FCP/E3T on environment generalization. I am also wondering how CEC would perform if combined with a teammate diversity method such as E3T.

**Other Comments Or Suggestions:**

- Typo on Line 161: “In contrast, we how…”
- Figure 3: the discussion of this figure on pg 4, right is very confusing because I’m not sure if the discussion refers to the right or left subfigure in Fig. 3. Can the authors also specify what the “Fixed Task” is in Fig. 3, left?

**Other Strengths And Weaknesses:**

- Strengths:

    - The paper is clear and well-written.

    - Statistical significance tests are provided

    - Core finding that environment randomization improves both task and cooperative generalization ability is interesting and somewhat surprising.

    - Method is evaluated with humans, and performs strongly

- Weaknesses:

    - Method is specific to overcooked, and cannot be applied out-of-the-box on other domains, without a procedural generator. Can the authors discuss what would be needed for their method to apply more broadly?

    - Method is extremely computionally expensive, requiring training for 3B steps, compared to conventional teammate generation methods.

    - No formal characterization of the relationship between environment layout diversity and strategic diversity in Overcooked, which might explain why increasing environment diversity leads to improved ZSC.

    - No explanation of why humans find CEC more enjoyable/adaptive compared to baseline agents

**Questions For Authors:**

1. How is the procedural environment generation method proposed in this paper different from domain randomization (Tobin et al. 2017)?

2. CEC is implemented using a self-play algorithm to isolate the effect of task diversity and partner diversity. Have the authors considered an algorithm that combines task and partner diversity?

3. Human-evaluation experiments (Q3): in these experiments, CEC falls short of E3T in cooperation with humans, but scores highest (by a small margin) according to human preferences. According to the analysis fo Figure 19, the authors state that this occurs because CEC is more adaptable to user behaviors, more strategically consistent, and better at avoiding collisions (Figure 11). However, the training process for CEC does not include any human data, so why would it learn any human norms?

4. Figure 13: why is it that the self-play score of each algorithm (on the diagonal) is no higher than any of the cross play scores, on the original 5 layouts? Typically, the algorithm self-play score is higher than algorithm cross-play scores.


Tobin, Josh, Rachel Fong, Alex Ray, Jonas Schneider, Wojciech Zaremba, and Pieter Abbeel. 2017. “Domain Randomization for Transferring Deep Neural Networks from Simulation to the Real World.” In *2017 IEEE/RSJ International Conference on Intelligent Robots and Systems (IROS)*, 23–30. [https://doi.org/10.1109/IROS.2017.8202133](https://doi.org/10.1109/IROS.2017.8202133).

**Relation To Broader Scientific Literature:**

The authors cite all relevant work that I am aware of, but they could improve the contextualization of their work w.r.t the UED literature, in particular, with respect to MAESTRO (Samvelyan et al. 2023) and Domain Randomization (Tobin et al. 2017) (already cited in the paper).

- MAESTRO: MAESTRO examines both environment and co-player autocurricula in 2p0s game settings, and is a prioritized-level-replay style method that leverages a randomized environment generator. How do the insights from MAESTRO relate to this paper. which addresses the fully cooperative setting?

- Domain Randomization: it’s not clear to me that the proposed environment randomization in this paper is any different from domain randomization.


The author should also discuss the MADRID paper (Samvelyan et al. 24), which also examines environment diversity as a way of exposing weaknesses in teammate policies.

Samvelyan, Mikayel, Davide Paglieri, Minqi Jiang, Jack Parker-Holder, and Tim Rocktäschel. 2024. “Multi-Agent Diagnostics for Robustness via Illuminated Diversity.” arXiv. [http://arxiv.org/abs/2401.13460](http://arxiv.org/abs/2401.13460).

**Theoretical Claims:**

N/A

---

> ### Author Rebuttal · Authors · 2025-04-01
>
> We thank the reviewer for their detailed feedback and recognition of our work’s novelty and clarity. Below, we address key concerns and how we plan to integrate this feedback:
> # Experiments Beyond Overcooked
> To help address your suggestions, we have conducted **3 new experiments showing that our method generalizes beyond Overcooked.** First, we would like to highlight the “Dual Destination” environment (Section 4, Figures 2-3 of the submission), where rewards are only given when both occupy separate green squares. For the rebuttal, we have conducted new experiments in a [partially observable version of this environment](https://rb.gy/iwg3wx), and found that CEC once again achieved high cross-play performance in novel environments (0.74), outperforming population-based methods (0.61) and naive self-play (0.03).
> # Combining Task and Partner Diversity
> Following your suggestion, we tested [combining partner diversity algorithms with environment diversity](https://rb.gy/g015me) using E3T under the CEC paradigm. Results showed CEC-E3T performed worse than other models on Overcooked’s five original layouts (CEC=130.51, CEC-E3T=28.21) but outperformed CEC-Finetune on 100 held-out grids (CEC-FT=41.73, CEC-E3T=58.13). The linked learning curves reveal that noisy partners combined with dynamic environments introduced additional noise, suggesting larger networks and longer training times for convergence may be needed compared to vanilla CEC.
> # New experiments without RNNs
> Per reviewer SmWr, we ablated RNNs in CEC (used for meta-learning per Wang et al, 2018; Rabinowitz et al, 2018). In Overcooked and Dual Destination, non-recurrent agents failed to converge or adapt. [Learning curves](https://rb.gy/nti7xf) show CEC with LSTMs converged successfully, while non-recurrent versions could not achieve positive rewards.
> # DR generates unsolvable environments
> We also appreciate the opportunity to clarify the distinction between CEC and standard domain randomization (DR). While CEC uses environment randomization, it employs a procedural generator to ensure all tasks are solvable, unlike naive randomization, which we found in our experiments often produces unsolvable layouts and poor learning signals. This mirrors the conclusions of the Overcooked Generalisation Challenge (Ruhdorfer et al., 2024), which shows that random/UED approaches often fail to generate solvable layouts or train agents capable of completing tasks. This highlights the need for structured procedural generation like ours.
> # Computational Feasibility of CEC
> We acknowledged that CEC’s 3 billion training timesteps exceed the computational budget of prior work like CoMeDi for a single layout. However, Population-based training (PBT) scales poorly when the goal is to cooperate on many levels: training eight agents (5M steps each) plus an ego cooperator (10M steps) results in a 50-million-step budget per layout. Scaling to 100 layouts would require 5 billion steps but still only generalize to fixed levels. CEC generalizes better to thousands of unseen layouts with lower total compute costs (Figures 3,6,9), making it more efficient for broad adaptability. We also note that for our experiments, we ensure that all algorithms are able to use the same compute budget of 3 billion steps, and we find that CEC is able to better use this compute to improve performance.
>
> We agree that **formalizing procedural generators** for naturalistic settings is a challenging direction for future work. One idea could involve combining program induction with procedural generation, as in WorldCoder (Tang et al., 2024), where LLMs generate OpenAI Gym-like environments with unknown dynamics. For CEC, inferred dynamics could be encoded into simulators like Unity before generating diverse scenes for RL training—a form of real-to-sim-to-real transfer (Torne et al., 2024).
> # CEC and related work
> Thank you for highlighting MAESTRO and MADRID, which we will include in our related works section. MAESTRO complements our work by focusing on zero-sum settings, and showing that in that case, jointly considering environment and partner diversity is optimal, while focusing on either leads to suboptimal outcomes. While we used uniform sampling over grids irrespective of the agent’s abilities during training, we will add to our discussion about how future work can explore autocurricula for partners and tasks to improve sample efficiency through strategies such as regret minimization ala MADRID and MAESTRO.
> # Analyzing CEC’s Norms
> The reviewer astutely questions how CEC learns human-like norms without explicit instructions. As noted in Section 6, we believe these norms reflect principles for cooperation in dynamic environments, such as “stay out of each other’s way.” These norms may be perceived as more human-like because humans also follow them.
>
> **Figures 13 and 14** show cross-play scores across all algorithms, forming the payoff matrix used to create Figure 8. The diagonal reflects results from Figure 6.

---

> > ### Comment · Reviewer_mZRU · 2025-04-05
> >
> > Thanks to the authors for providing a comprehensive rebuttal, and apologies for the late response.
> >
> > The additional results on the Dual Destination environment somewhat address my concerns about the generalizability of the approach to other domains, but not fully, as the dual destination problem was designed to illustrate the authors' point. It would be more convincing if the authors present additional experiments on another actual AHT domain such as Hanabi.
> >
> > The main argument that's not convincing to me is the argument on the computational feasibility of CEC. The authors argue that CEC is a more efficient approach than population-based AHT methods when it comes to training a policy that can deal with 100 layouts, because if we train a population-based AHT method separately for each layout with a population size of 8, it would take 5 billion timesteps total (compared to 3B for CEC). This argument has a couple holes: (1) the numbers were specifically chosen to make CEC more efficient -- if the population-based AHT method was trained with a population size of 4, then the PBT method would only need a computational budget of 2.5B steps; (2) it's unreasonable to assume that we would start training the policy from scratch for each layout.
> >
> > The two issues I pointed out above are somewhat pedantic though, and I do not view computational feasibility as a key issue for the paper. In my opinion, the key contribution of this paper/CEC is demonstrating that cross-task generalization and cross-team generalization (AHT) are not orthogonal axes, which is an important finding alone. However, the paper reads to me as though it is presenting CEC as a novel AHT algorithm. This actually weakens the impact of the main contribution, since characterizing CEC as an AHT algorithm is challenging --- as the argument on computational feasibility of CEC vs AHT demonstrates, constructing a totally fair comparison between CEC and existing AHT methods is difficult. I encourage the authors to discuss limitations of the comparison between CEC and AHT methods in future iterations of the paper.

---

> > > ### Author Response · Authors · 2025-04-08
> > >
> > > Thank you for engaging in the rebuttal process and providing a more nuanced perspective on CEC as an AHT algorithm. We agree that in our example we tried to illustrate how with a large population trained from scratch on each new environment, CEC proves to be more efficient in cross-partner cross-environment generalization than population based methods. Due to the current limitations of PBT methods, namely that it is hard to guarantee diversity in partner strategies if the only variation between populations of self-play agents is the random initialization or network architecture, a larger population size increases the likelihood of new strategies being found during training, making the ego cooperator more robust to novel partners at test-time. For instance, in continuous domains you might require a very large population to obtain sufficient strategy coverage, since each policy might diverge by a very small amount but be semantically similar. However, should there be a method for accurately estimating differences between partner strategies, then the reviewer correctly points out that PBT methods can be more sample efficient, albeit if they are trained from scratch on every new environment they will likely prove to be more computationally expensive than CEC. For instance, if we use the reviewer’s numbers of training a population of size 4 and ego cooperator for 25 million steps per environment, retraining an agent for each of 120 environments would take 3 billion steps, but would lack the generality to thousands of levels that CEC boasts without any additional training.
> > >
> > > If we assume we are not going to retrain a PBT method from scratch when exposed to a new level, this becomes murkier territory to evaluate the differences between CEC and PBT. As we demonstrate in Figures 6 and 9 of our paper, the CEC-Finetune experiments show sequentially training on a set of levels (first learn on set of levels A, then specialize on level B) leads to catastrophic forgetting in agents if they try to play levels in distribution of the set of levels A. As evidenced from the continual learning literature, there is a fundamental tradeoff between stability and plasticity (Mermillod et al, 2013; Kim et al, 2023; Elsayed and Mahmood, 2024) when trying to deal with sequential learning tasks, a problem we showed to persist in Figures 6 and 9 (high plasticity but low stability). An alternative to sequentially learning tasks with partner diversity methods is to do both forms of learning simultaneously, that is, combine partner and environment diversity. However, as depicted in our additional experiments, this is a non-trivial problem to solve, since naive combinations of partner and environment diversity methods lead to a [highly noisy training procedure](https://rb.gy/g015me) where agents fail to learn or generalize effectively (CEC=130.51, CEC-E3T=28.21).
> > >
> > > Based on the experiments we conducted and our understanding of the literature, which we acknowledge may have limitations, we observed that retraining a PBT method on every level can be less efficient than CEC when aiming to ensure that the ego cooperator is robust to new partners on new environments. Specifically, increasing the number of partners in a population improves robustness but also increases inefficiency in PBT methods. We appreciate the reviewer’s perspective that integrating population-based methods with cross-environment training can blur the distinction between the two approaches. In response, we will incorporate their suggestion to reframe our contributions as a more detailed exploration of how cross-task and cross-partner generalization can complement each other rather than being mutually exclusive.
> > >
> > > Ref:
> > >
> > > Elsayed, Mohamed, and A. Rupam Mahmood. "Addressing loss of plasticity and catastrophic forgetting in
> > > continual learning." arXiv preprint arXiv:2404.00781 (2024).
> > >
> > > Kim, Sanghwan, et al. "Achieving a better stability-plasticity trade-off via auxiliary networks in continual learning." Proceedings of the IEEE/CVF Conference on Computer Vision and Pattern Recognition. 2023.
> > >
> > > Mermillod, Martial, Aurélia Bugaiska, and Patrick Bonin. "The stability-plasticity dilemma: Investigating the continuum from catastrophic forgetting to age-limited learning effects." Frontiers in psychology 4 (2013): 504.

---

### Decision · Program_Chairs · 2025-05-01

**Decision:**

Accept (oral)

**Comment:**

After engaging in discussions with the authors, all reviewers feel the paper should be accepted to ICML.  The authors are strongly encouraged to find a way to add the additional results provided in the discussion phase to the paper -- reviewers were quite happy with them.